# Shielded bifunctional nanoreactor enabled tandem catalysis for plasma methane coupling

Chunqiang Lu[1,2,7], Yaolin Wang[3,7], Dong Tian[1,2,7], Ruidong Xu[1,2], Roong Jien Wong [4,5], Shibo Xi[5], Wen Liu[4], Hua Wang [1,2] ✉, Xin Tu [3] ✉ & Kongzhai Li [1,2,6] ✉

The direct conversion of methane into valuable unsaturated $C_2$ hydrocarbons ($C_2H_2$ and $C_2H_4$) attracts growing attention. Non-thermal plasma offers a promising approach for this process under mild conditions. However, the competing formation of $C_2H_6$ and excessive dehydrogenation limit the selectivity toward $C_2H_2$ and $C_2H_4$. Herein, we develop a promising shielded bifunctional nanoreactor with a hollow structure and mesoporous channels ($Na_2WO_4$-$Mn_3O_4$/m-$SiO_2$) that effectively limits $CH_4$ overactivation and promotes selective coupling to form $C_2H_2$ and $C_2H_4$ under plasma activation, achieving 39% $CH_4$ conversion with 42.3% $C_2H_2$ and $C_2H_4$ fraction. This nanoreactor features isolated $Na_2WO_4$ embedded within the channels and $Mn_3O_4$ confined in the cavity of the $SiO_2$ hollow nanospheres, enabling internal tandem catalysis at co-located active sites. $Na_2WO_4$ induces the conversion of diffused $CH_4$ and $CH_3$ into reactive intermediates (˙CH and ˙$CH_2$), which subsequently couple on the $Mn_3O_4$ surface to form $C_2H_2$ and $C_2H_4$. Furthermore, the mesoporous channels inhibit the plasma discharge within the nanoreactor, preventing deep dehydrogenation of $CH_x$ species to solid carbon. This nanoreactor demonstrates a highly selective route for the nonoxidative conversion of methane to valuable $C_2$ hydrocarbons, offering a new paradigm for the rational design of catalysts for plasma-driven chemical processes.

Unsaturated $C_2$ hydrocarbons, crucial building blocks in the chemical industry, are primarily produced from the energy-intensive processing of crude oil. The development of sustainable alternatives via gas conversion has attracted increasing interest[1,2]. While conventional gas conversion methods rely on indirect syngas routes, direct methane conversion to unsaturated $C_2$ hydrocarbons offers a more attractive pathway[3,4]. Direct conversion of $CH_4$ to value-added fuels and chemicals can be achieved through either nonoxidative or oxidative catalytic routes[5–7]. The nonoxidative pathway offers high $C_2$ selectivity and atom utilization efficiency but demands endothermic conditions (up to 1100 °C), incurring substantial energy costs and $CO_2$ emissions[1]. In contrast, oxidative coupling of methane (OCM) can significantly lower

[1]State Key Laboratory of Complex Nonferrous Metal Resources Clean Utilization, Kunming University of Science and Technology, Kunming 650093, P. R. China. [2]Faculty of Metallurgical and Energy Engineering, Kunming University of Science and Technology, Kunming 650093, P. R. China. [3]Department of Electrical Engineering and Electronics, University of Liverpool, Liverpool L69 3GJ, UK. [4]School of Chemistry, Chemical Engineering and Biotechnology, Nanyang Technological University, 62 Nanyang Drive, Singapore 637459, Singapore. [5]Institute of Sustainability for Chemicals, Energy and Environment (ISCE2), Agency for Science, Technology and Research (A*STAR), 1 Pesek Road, Jurong Island, Singapore 627833, Republic of Singapore. [6]Southwest United Graduate School, Kunming 650092, P. R. China. [7]These authors contributed equally: Chunqiang Lu, Yaolin Wang, Dong Tian. ✉e-mail: wanghua65@163.com; xin.tu@liv.ac.uk; kongzhai.li@foxmail.com

the reaction temperature to ~600 °C in the presence of $O_2$[8,9]. However, this method suffers from overoxidation of $CH_4$ to thermodynamically favored CO and $CO_2$, limiting the yield of unsaturated $C_2$ hydrocarbons[8,10]. These challenges highlight the urgent need for innovative strategies to enhance methane activation and coupling efficiency.

Non-thermal plasma (NTP) offers a promising solution for the homolytic activation of $CH_4$ into radicals, enabling nonoxidative coupling of methane (NOCM) under mild conditions[11,12]. Among NTP techniques, dielectric barrier discharge (DBD) has been extensively explored for methane coupling. However, the main challenge is the limited control over product distribution, particularly in achieving selective production of unsaturated $C_2$ hydrocarbons, such as acetylene ($C_2H_2$) and ethylene ($C_2H_4$). Instead, ethane ($C_2H_6$) is predominantly produced in NOCM using DBD reactors[11,13–16]. This selectivity issue originates from the significantly longer lifetime of $CH_3$ radicals (>1 ms) compared to $CH_2$ (<30 ns) and CH (<5 ns) radicals, which favors undesired reaction pathways[17]. Integrating tailored catalysts into DBD reactors has great potential to enhance selectivity toward targeted products[14,18,19]. Ideally, such catalysts would regulate the dehydrogenation process and selectively stabilize CH and $CH_2$ intermediates, directing the reaction toward the formation of $C_2H_2$ and $C_2H_4$.

The plasma-catalytic NOCM process involves heterogeneous surface reactions, highlighting the importance of the rational design of catalysts with tailored active sites and morphologies to maximize performance. In conventional OCM, $Na_2WO_4$ and $Mn_2O_3$ have proven effective in $CH_4$ activation and C–C coupling[20]. Specifically, $Na_2WO_4$ generates reactive oxygen species to activate $CH_4$, while $Mn_2O_3$-supported O species promote the coupling of $^*CH_3$ intermediates[21–23]. Given their critical roles in methane coupling, integrating $Na_2WO_4$ and $Mn_2O_3$ into plasma-catalytic NOCM offers a promising avenue to enhance the production of unsaturated $C_2$ hydrocarbons through their synergistic effects – a strategy that remains underexplored. Notably, $Na_2WO_4$ and $Mn_2O_3$ function at distinct stages of the methane coupling pathway. Plasma pre-activates $CH_4$, facilitating $Na_2WO_4$-mediated enhancement of $CH_x$ dehydrogenation into $^*CH$ and $^*CH_2$ surface species, while $Mn_2O_3$ promotes subsequent coupling of these intermediates to form $C_2H_2$ and $C_2H_4$. Inspired by this synergistic interplay, we propose that a tandem plasma-catalysis system using spatially structured $Na_2WO_4$-$MnO_x$ catalysts—designed to sequentially optimize dehydrogenation and coupling steps—could potentially achieve enhanced efficiency in $C_2$ hydrocarbon formation compared to plasma-catalysis systems using randomly structured catalysts.

To address the challenges of overactivation and selectivity control in plasma-assisted methane coupling, we propose a promising assembled nanoreactor ($Na_2WO_4$-$Mn_3O_4$/m-$SiO_2$, denoted as WMO/m-$SiO_2$) designed for integration into a DBD plasma reactor (Supplementary Fig. 1). This nanoreactor features a hollow structure with mesoporous channels, accessibility to reactants both internally and externally. This design provides a shielding effect for $CH_4$ molecules within the channels, preventing excessive dehydrogenation by plasma-generated reactive species. Furthermore, the WMO/m-$SiO_2$ reactor demonstrates tandem catalytic functionality, significantly enhancing selectivity toward target $C_2$ products. Specifically, $Na_2WO_4$ located within the m-$SiO_2$ channels initiates $CH_4$ activation, generating $^*CH_2$ and $^*CH$ surface intermediates. Concurrently, $Mn_3O_4$ species confined within the cavity of the $SiO_2$ spheres promote facile coupling of these CH and $CH_2$ intermediates to form $C_2H_2$ and $C_2H_4$. Our study demonstrates significantly enhanced $C_2H_2$ and $C_2H_4$ yields compared to conventional catalysts under analogous DBD plasma conditions. This work offers a strategy to overcome critical limitations in plasma-catalysis, advancing efficient methane conversion to high-value $C_2$ hydrocarbons.

## Results

### Structural characterization of the catalysts

The synthesized mesoporous $SiO_2$ (m-$SiO_2$) featured interconnected nanospheres with diameters ranging from 95 to 135 nm (Fig. 1a and Supplementary Fig. 2a). Each nanosphere exhibited a hollow structure with a cavity diameter of ~65 nm and a shell thickness of ~20 nm (Supplementary Fig. 2b), along with a relatively high specific surface area of 240 $m^2 g^{-1}$ (Supplementary Table 1). Loading manganese and tungsten species onto m-$SiO_2$ did not affect the hollow structure, and the resulting WMO/m-$SiO_2$ nanoreactor retained surface porosity with particles localized both within the cavity and on the external surface (Fig. 1b, d and Supplementary Fig. 3). X-ray diffraction (XRD) confirmed the presence of $Mn_3O_4$ and amorphous $SiO_2$ in the WMO/ m-$SiO_2$ composite (Supplementary Fig. 4). High-resolution transmission electron microscopy (HRTEM) revealed that the dispersed particles on the shell and cavity walls of m-$SiO_2$ were $Na_2WO_4$ and $Mn_3O_4$, respectively (Fig. 1d–f and Supplementary Fig. 3). The measured lattice spacings of 0.38 and 0.25 nm corresponded to $Na_2WO_4$ (111) and $Mn_3O_4$ (211), respectively (Fig. 1e). Notably, although XRD confirmed the presence of $Mn_3O_4$ (Supplementary Fig. 4), scanning electron microscopy (SEM) elemental distribution mapping (Supplementary Figs. 5, 6) of WMO/m-$SiO_2$ revealed negligible Mn and W signals compared to WMO/$SiO_2$ and WMO/ZSM-5 (Zeolite Socony Mobil-5). Unless otherwise specified, WMO/m-$SiO_2$, WMO/$SiO_2$, and WMO/ZSM-5 refer to samples with 1% $Na_2WO_4$-5% Mn loaded on the support material. These findings indicate that $Na_2WO_4$ and $Mn_3O_4$ particles are predominantly encapsulated within the m-$SiO_2$ spheres. Additionally, energy-dispersive X-ray spectroscopy (EDS) scans in Fig. 1f further confirmed the presence of Mn species inside the m-$SiO_2$ cavity.

### Plasma-catalytic NOCM reaction

We evaluated $CH_4$ conversion under three different conditions: plasma-only (no catalyst, no external heating), catalysis-only (external heating at 250 °C, no plasma), and plasma-catalysis (coupled plasma and catalysts, no external heating). Under plasma-only and plasma-catalysis conditions, the measured temperature was ~250 °C. In plasma-only mode, $C_2$-$C_3$ hydrocarbons dominated, with a maximum production of 31.2 μmol min$^{-1}$ at a $CH_4$ conversion of 33% (Supplementary Figs. 7, 8). Hydrocarbons with four or more carbon atoms were excluded due to negligible concentrations (<5% relative selectivity compared to $C_2$ products). Figure 1c shows the distributions of $C_2H_4$ and $C_2H_2$ within the $C_2$-$C_3$ range. A synchronized increase in $C_2H_2$ and $C_2H_4$ proportions was observed, which can be attributed to the closely aligned energetic thresholds of electron-induced $CH_4$ conversion into $CH_2$ and CH species[13]. WMO/m-$SiO_2$ exhibited no catalytic activity for methane conversion at 250 °C in the absence of plasma. However, under NTP conditions with WMO/m-$SiO_2$, the production of $C_2H_4$ and $C_2H_2$ significantly increased to 30.3 μmol g$^{-1}$ min$^{-1}$ (12.8 μmol g$^{-1}$ min$^{-1}$ for $C_2H_4$ and 17.5 μmol g$^{-1}$ min$^{-1}$ for $C_2H_2$), surpassing WMO/$SiO_2$ and WMO/ZSM-5 by factors of ~5 and 3.4, respectively (Fig. 1c). With the WMO/m-$SiO_2$ nanoreactor, the proportion of unsaturated hydrocarbons in the $C_2$–$C_3$ range increased significantly from 17.7% (plasma-only) to 42.3% (Fig. 1c and Supplementary Fig. 9). Simultaneously, the total yield of $C_2H_4$ increased markedly from 2.6 to 6.4 μmol min$^{-1}$, while $C_2H_2$ increased from 3.1 to 8.8 μmol min$^{-1}$. Notably, all three catalysts selectively promoted $C_2H_4$ and $C_2H_2$ production via deep dehydrogenation and coupling of $CH_4$, rather than promoting overall methane conversion (Fig. 1h).

The pore sizes of m-$SiO_2$ and WMO/m-$SiO_2$ ranged from 20 to 40 nm, distinct from those of WMO/$SiO_2$ and WMO/ZSM-5 (Fig. 1g and Supplementary Fig. 10). As shown in Supplementary Table 1, WMO/ ZSM-5 exhibited the highest surface area of 226 $m^2 g^{-1}$, significantly exceeding that of WMO/m-$SiO_2$ (67 $m^2 g^{-1}$). However, plasma-only and

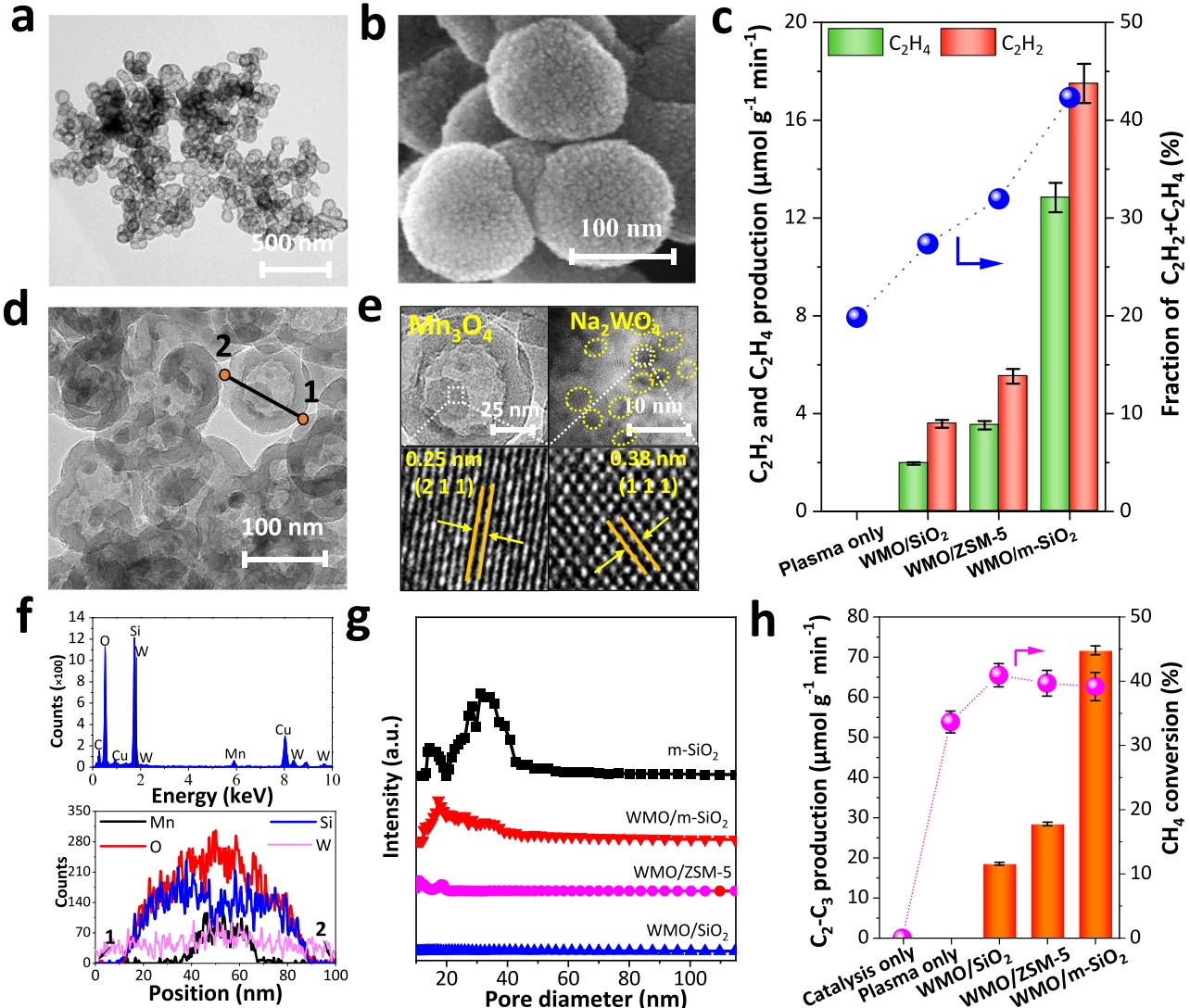

**Fig. 1 | Characterization and catalytic performance. a** TEM image and particle size distribution of m-SiO₂. **b** SEM image of WMO/m-SiO₂. **c** Production rate and molar fraction of C₂H₄ and C₂H₂ within C₂-C₃ hydrocarbons (Conditions: 1 bar, specific energy input (SEI) 5.1 kJ L⁻¹, where SEI is defined as plasma discharge power divided by the gas flow rate; Feed gas 5 vol% CH₄/Ar, total flow rate 200 mL min⁻¹, discharge power 17 W, experiment duration 60 min). Error bars (standard deviation) in the figure were obtained from three sampling runs. **d, e** TEM and HRTEM images of WMO/m-SiO₂. **f** EDS line scans (from Point 1 to Point 2) from Fig. 1d. **g** Pore size distributions of m-SiO₂, WMO/m-SiO₂, WMO/ZSM-5, and WMO/SiO₂. **h** Production rate of C₂-C₃ hydrocarbons and methane conversion for catalysis-only, plasma-only and plasma-catalysis systems.

plasma-catalysis conditions (using WMO/ZSM-5, WMO/SiO₂ and WMO/m-SiO₂) exhibited similar discharge properties (Supplementary Figs. 11–13). Additionally, packing the discharge zone with silica supports yielded consistent C₂H₂ and C₂H₄ levels (Supplementary Fig. 14). Furthermore, despite its lower surface area, WMO/m-SiO₂ demonstrated better CH₄ coupling efficiency to C₂H₂ and C₂H₄, highlighting the critical role of its unique hollow mesoporous structure in enhancing the synergistic interaction between NTP and catalysis.

**Evaluation of the effect of catalyst position on performance**

TEM and scanning transmission electron microscopy (STEM) images depict the morphology of the catalysts and the location of metal oxide particles, respectively (Fig. 2a and Supplementary Fig. 15). The metal oxide particles were selectively deposited in three configurations: (1) exclusively inside m-SiO₂ (In-m-SiO₂), (2) partially distributed within m-SiO₂ (Both-m-SiO₂), and (3) predominantly deposited on the exterior of m-SiO₂ (Out-m-SiO₂). For both "In-m-SiO₂" and "Both-m-SiO₂", the Mn₃O₄ sizes ranged from 5 to 35 nm (Supplementary Fig. 15e, f).

TEM analysis of Both-m-SiO₂ revealed a significant reduction in the Si signal around Particle 1 (50–90 nm), while no similar decrease was observed near Particle 2 (150–200 nm) (Fig. 2a). Moreover, the Mn signal intensified in both particle types, suggesting that Particle 1 and Particle 2 are located on the exterior and interior surfaces of the m-SiO₂ nanosphere, respectively (Fig. 2a). XRD patterns further confirmed the spatial distribution of Na₂WO₄ and Mn₃O₄ particles. Samples with higher diffraction intensities and more pronounced peaks corresponded to increased exposure of Mn₃O₄ species on the external surface of m-SiO₂ (Supplementary Fig. 16).

EDS analysis of Particle 1 (internal) revealed weaker carbon and stronger oxygen signals compared to Particle 2 (external) (Fig. 2a and Supplementary Fig. 17). This suggests that m-SiO₂ protects Mn₃O₄ particles from direct exposure to CH₄ plasma. This shielding effect arises from the "Debye shielding" mechanism, where plasma discharge cannot penetrate the mesopores of m-SiO₂ due to their pore diameters being smaller than the Debye length (typically hundreds of nanometers)[19,24,25]. Thus, the shielded internal cavity prevents the

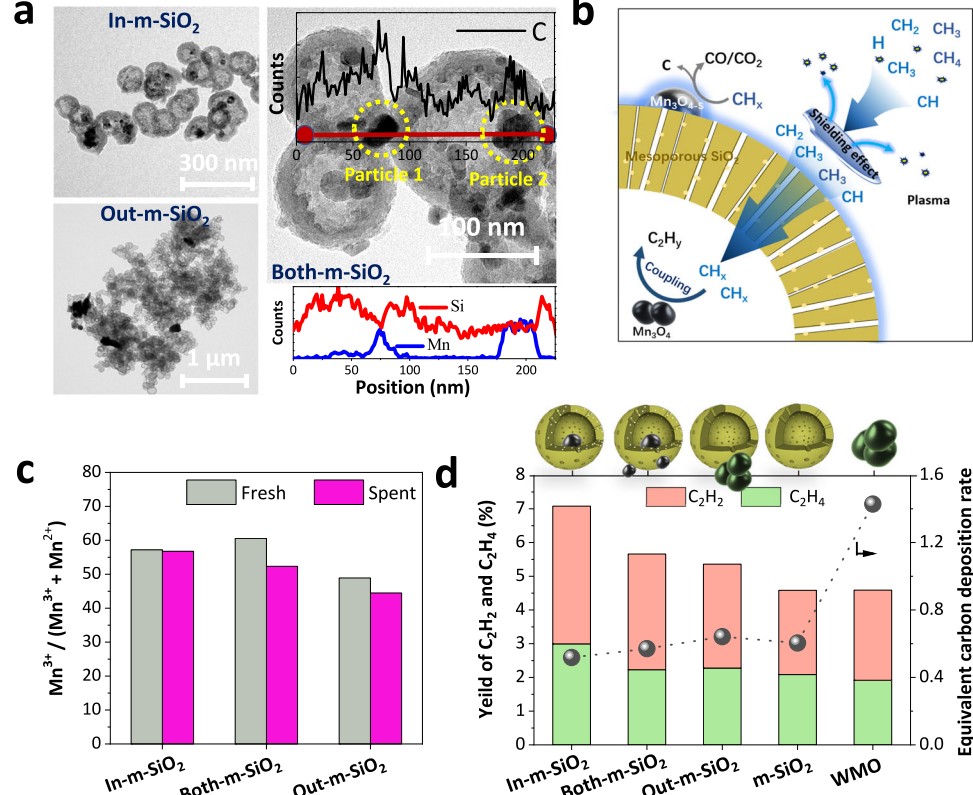

**Fig. 2 | Effect of catalyst location m-SiO₂ on NTP-CM performance. a** TEM image and EDS spectra of spent catalysts In-m-SiO₂, Out-m-SiO₂, and Both-m-SiO₂ (after 60 min of reaction). **b** Schematic illustration of the role of m-SiO₂ and the effect of catalyst position on methane coupling. **c** $Mn^{3+}/(Mn^{3+}+Mn^{2+})$ ratio for fresh and spent catalysts. **d** Yields of $C_2H_2$ and $C_2H_4$ (defined as the product of CH₄ conversion and the selectivity of $C_2H_2$ and $C_2H_4$) and equivalent carbon deposition rate (ECR, defined as the solid carbon selectivity on the catalyst divided by the methane conversion) (Conditions: 1 bar, SEI 5.1 kJ L⁻¹, total flow rate 200 mL min⁻¹, discharge power 17 W, experiment duration 60 min).

reduction of Mn₃O₄ particles and mitigates direct carbon deposition. This hypothesis is further supported by the observed decrease in the $Mn^{3+}/(Mn^{3+}+Mn^{2+})$ ratio as more manganese oxide particles are located outside the m-SiO₂ layer (Fig. 2c and Supplementary Fig. 18). Previous studies have shown that low-valent metal catalysts (oxides, carbides, or metals) are highly effective for the deep dehydrogenation of CH₄[26,27]. Consistent with these findings, Supplementary Fig. 19 shows that Mn₃O₄ particles exhibited a stronger carbon signal than the m-SiO₂ surface. Furthermore, as Na₂WO₄ and Mn₃O₄ particles are encapsulated within m-SiO₂ nanospheres, the yield of $C_2H_2$ and $C_2H_4$ decreased from 7.0% (WMO/m-SiO₂) to 4.5% (Na₂WO₄·Mn₃O₄, denoted as WMO). This trend implies that converting an equivalent amount of methane leads to more carbon deposition when fewer encapsulated Na₂WO₄ and Mn₃O₄ particles are present (Fig. 2d and Supplementary Fig. 20).

### The role of Na₂WO₄ and Mn₃O₄ in plasma-catalytic NOCM reaction

The 1% Na₂WO₄-5% Mn₃O₄/m-SiO₂ catalyst achieved a combined $C_2H_4$ and $C_2H_2$ selectivity of 18.1%, surpassing that of Na₂WO₄/m-SiO₂ (14.5%), 5% Mn₃O₄/m-SiO₂ (13.2%) and plasma-only conditions (7.9%) (Fig. 3a and Supplementary Fig. 21). A comparison of CH₄ conversion and C₂-C₃ hydrocarbon distribution in the DBD reactor with previous studies is provided in Supplementary Table 2. The 1% Na₂WO₄-5% Mn₃O₄/m-SiO₂ (WMO/m-SiO₂) catalyst demonstrated the high selectivity for $C_2H_2$ and $C_2H_4$, while maintaining competitive methane conversion. Among reported studies, this work achieved a lower energy cost (EC) for CH₄ conversion (6.8 MJ/mol), demonstrating the effectiveness of the catalyst in plasma-catalytic NOCM reaction.

Notably, the catalyst exhibited stable performance for over 25 h (Supplementary Figs. 22, 23). X-ray photoelectron spectroscopy (XPS) and XRD analyses confirmed that the dominant oxidation states of tungsten and manganese species in WMO/m-SiO₂, as well as in 5% Mn/m-SiO₂ and 1% Na₂WO₄/m-SiO₂, remained unchanged after the reaction (Supplementary Figs. 24–26). Supplementary Fig. 21 shows that increasing the Na₂WO₄ loading from 0.5 to 5% increased carbon deposition from 16.4 to 24.6%. In contrast, higher Mn₃O₄ content reduced carbon deposition. These findings suggest that Na₂WO₄/m-SiO₂ more effectively promotes the further dehydrogenation of methane compared to Mn₃O₄/m-SiO₂.

The involvement of catalyst-bound oxygen species is well established in methane dehydrogenation[5,9] and the coupling of intermediate species[28–30], highlighting their crucial roles in these reactions. To investigate this effect, we pretreated WMO/m-SiO₂ with H₂ at 450 °C for different durations, generating a series of WMO/m-SiO₂ samples with varying oxygen contents. H₂ pretreatment at 450 °C primarily reduced Mn₃O₄ but not tungsten species, as evidenced by H₂ temperature-programmed reduction (H₂-TPR) and XPS (Fig. 3c and Supplementary Figs. 27, 28). H₂ treatment resulted in increased carbon deposition and decreased selectivity for $C_2H_4$ and $C_2H_2$ when the WMO/m-SiO₂ nanoreactor was reduced for 5–20 min, accompanied by a decline in $Mn^{3+}$ content and oxygen loading within the catalyst (Fig. 3b, c). These observations align with the trend that increasing the Mn content (β) from 2 to 10% in 1% Na₂WO₄-β Mn₃O₄/m-SiO₂ slightly enhanced the production of C₂-C₃ hydrocarbons (Supplementary Fig. 29). These findings suggest that MnOₓ acts as an oxygen carrier, promoting the coupling of active species (CHₓ and H) and reducing methane cracking. This highlights

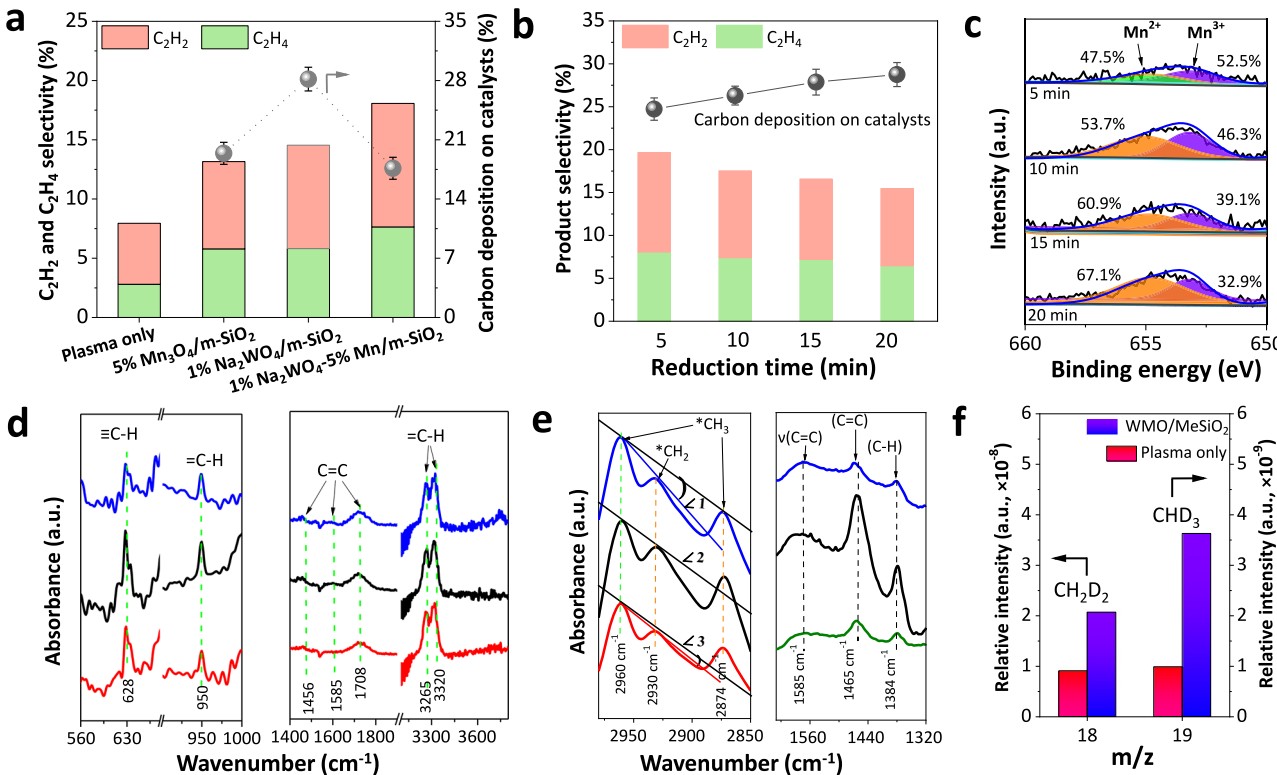

**Fig. 3 | Performance of Mn and W species. a** Selectivity of products (carbon deposited on the catalyst, $C_2H_4$ and $C_2H_2$) for plasma-only and plasma-catalysis systems (Conditions: 1 bar, SEI 5.1 kJ $L^{-1}$, total flow rate 200 mL $min^{-1}$, discharge power 17 W, experiment duration 60 min). Error bars (standard deviation) in the figure were obtained from three sampling runs. **b** Selectivity of carbon deposited on the catalyst and selectivity of $C_2H_4$ and $C_2H_2$ for reduced WMO/m-SiO$_2$ (reduced at 450 °C with H$_2$). **c** Mn $2p$ XPS spectra of reduced WMO/m-SiO$_2$ (pretreated with H$_2$ for 5, 10, 20, and 25 min). **d** IR spectra of WMO/m-SiO$_2$ (black), Mn$_3$O$_4$/m-SiO$_2$ (blue), and Na$_2$WO$_4$/m-SiO$_2$ (red) under plasma-catalysis conditions. **e** Quasi-in situ DRIFT spectra of catalysts after plasma-catalytic NOCM reaction. **f** CH$_2$D$_2$ and C$_2$HD$_3$ species generated under plasma-only and plasma-catalysis (with WMO/m-SiO$_2$) conditions (feed gas 2.5 vol% CH$_4$-2.5 vol% CD$_4$/Ar, SEI 5.1 kJ $L^{-1}$, total flow rate 200 mL min$^{-1}$; experiment duration 60 min).

the importance of optimizing oxygen content in catalysts to maximize performance.

## CH$_x$ species absorbed on the catalyst and in the gas phase

In situ plasma-coupled Fourier transform infrared (FTIR) spectroscopy was used to investigate plasma-assisted surface reactions on WMO/m-SiO$_2$, 5% Mn$_3$O$_4$/m-SiO$_2$ and 1% Na$_2$WO$_4$/m-SiO$_2$. As shown in Supplementary Fig. 30, the intensities of the IR peaks at 628 cm$^{-1}$ (≡C-H) and 950 cm$^{-1}$ (=C-H) decreased as the reaction progressed. These absorbed ≡C-H and =C-H bands are associated with key intermediates involved in the formation of $C_2H_2$ and $C_2H_4$ during the catalytic process[31]. Additional IR bands at 1465, 1585, 1708, 3265, and 3320 cm$^{-1}$ correspond to C = C stretching vibrations on the catalyst surface (Fig. 3d)[31–33]. Notably, WMO/m-SiO$_2$ exhibited the highest intensities for peaks associated with ≡C-H, =C-H, and C=C compared to catalysts containing only Mn or W. This finding suggests that the synergistic interaction between Mn$_3$O$_4$ and Na$_2$WO$_4$ sites effectively promotes the formation of surface-adsorbed *CH and *CH$_2$ groups, ultimately enhancing the yield of $C_2H_4$ and $C_2H_2$.

Quasi-in situ DRIFTS characterization provided further insights into the types of adsorbed species remaining on the catalyst surface after the reaction (Supplementary Figs. 31, 32). As shown in Fig. 3e, several key peaks were observed, including C−H stretching from *CH$_3$ (2960 and 2872 cm$^{-1}$)[34], C-H stretching from *CH$_2$ (2930 cm$^{-1}$)[34], C=C bonds (1585 and 1465 cm$^{-1}$)[32,33] and C-H bond bending/deformation modes (1384 cm$^{-1}$)[35]. In this study, the angles (∠1 = 12°, ∠2 = 0°, and ∠3 = 3°) between the standard slope and tangents (peak 2960 to peak 2930 cm$^{-1}$) were used to measure the relative proportions of absorbed

*CH$_3$ and *CH$_2$ species on the catalyst surface. These species serve as precursors for the formation of ethane and ethylene, respectively[36]. Among the catalysts, 1% Na$_2$WO$_4$-5% Mn$_3$O$_4$/m-SiO$_2$ exhibited the highest *CH$_2$ intensity, followed by 1% Na$_2$WO$_4$/m-SiO$_2$ and 5% Mn/m-SiO$_2$. Notably, compared to 1% Na$_2$WO$_4$/m-SiO$_2$, 5% Mn/m-SiO$_2$ promoted the formation of surface C=C (1585 and 1465 cm$^{-1}$), indicating that *CH$_2$ formation predominantly occurred at W sites, while Mn sites facilitated the coupling of CH$_2$ to form C=C bonds. Although the internal Mn$_3$O$_4$ sites are not directly exposed to plasma (Supplementary Fig. 3), varying the Mn$_3$O$_4$ loading within the m-SiO$_2$ spheres led to observable changes in product distribution and the relative intensity of adsorbed *CH$_3$ and *CH$_2$ species (Supplementary Figs. 21, 31). This suggests that CH$_x$ radicals can diffuse or migrate at least 20 nm to reach Mn$_3$O$_4$ sites within their lifetime, enabling them to access the interior of the catalyst for subsequent reactions.

## Methane isotopic labeling experiments

Methane isotopic labeling experiments, conducted using a parallel flow of CH$_4$ and CD$_4$, revealed an increase in CH$_3$ and CH$_2$ radicals during the plasma-catalyzed reaction over WMO/m-SiO$_2$ compared to the plasma-only condition (Fig. 3f). Detailed experimental procedures are provided in the Methods section, and the corresponding conversions of CD$_4$ and CH$_4$, along with product distributions, are shown in Supplementary Fig. 33. The experiment was designed to probe the complex dynamics within the discharge field, where multiple collisions and coupling reactions occur, leading to the reversible activation of C-H bonds, facilitating the reformation of nascent methane from activated C$_x$H$_y$ intermediates[36]. This behavior contrasts with the

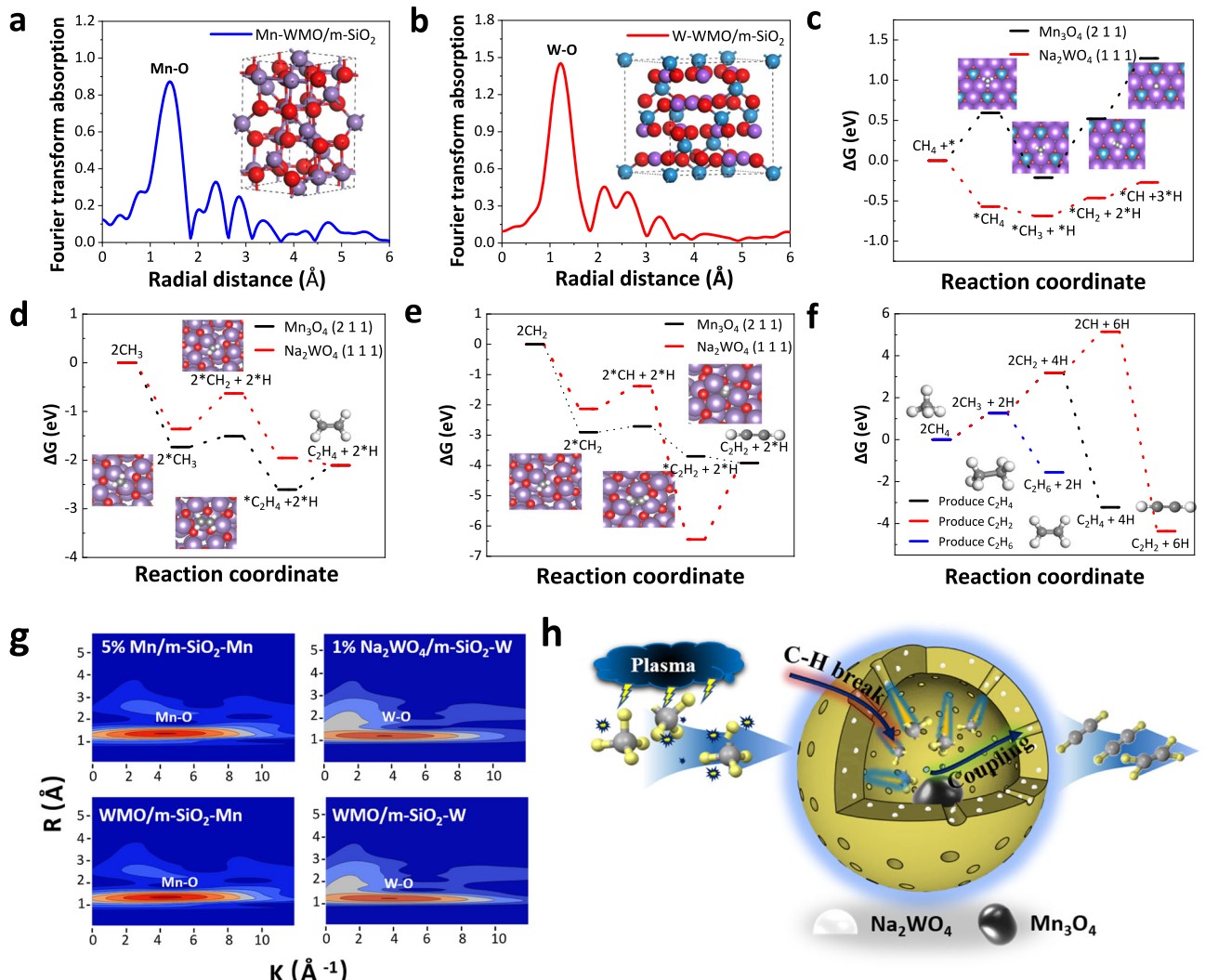

**Fig. 4 | The structures of WMO/m-SiO$_2$ and DFT calculations. a** Fourier transform (FT) of the EXAFS spectrum of Mn and the DFT-optimized structure of Mn$_3$O$_4$ on WMO/m-SiO$_2$. **b** FT-EXAFS spectra of W and DFT-optimized structure of Na$_2$WO$_4$ on WMO/m-SiO$_2$. **c** DFT-optimized geometries of ˙CH$_4$ dehydrogenation to ˙CH$_3$, ˙CH$_2$, and ˙CH on Mn$_3$O$_4$ (211) and Na$_2$WO$_4$ (111) surfaces. **d** DFT-optimized geometries of ˙CH$_3$ to C$_2$H$_4$ on Mn$_3$O$_4$ (211) and Na$_2$WO$_4$ (111) surfaces. **e** DFT-optimized geometries of ˙CH$_2$ coupled to C$_2$H$_2$ on Mn$_3$O$_4$ (211) and Na$_2$WO$_4$ (111) surfaces. **f** DFT-optimized geometries of intermediates in the plasma-catalytic conversion of CH$_4$ to radicals, C$_2$H$_2$, C$_2$H$_4$, and C$_2$H$_6$. **g** Wavelet transform plots of the Mn K-edge and W K-edge. **h** Schematic illustration of C$_2$H$_2$ and C$_2$H$_4$ formation pathways on WMO/m-SiO$_2$.

essentially irreversible C-H activation in traditional OCM[29]. The detection of mixed CH$_2$D$_2$ and CHD$_3$ isotopes during plasma-catalyzed NOCM on WMO/m-SiO$_2$ confirmed this mechanism. Compared to the plasma-only system, the plasma-catalytic system with WMO/m-SiO$_2$ demonstrated significantly higher production of CH$_2$D$_2$ and CHD$_3$ (Fig. 3f). This result suggests that WMO/m-SiO$_2$ enhances the activation of CH$_4$ and the generation of CH$_3$/CD$_3$ and CH$_2$/CD$_2$ radicals. The enriched pool of CH$_2$ species ultimately promotes dimerization into C$_2$H$_2$ and C$_2$H$_4$[11]. In summary, the synergistic interaction between Mn$_3$O$_4$ and Na$_2$WO$_4$ sites in the WMO/m-SiO$_2$ nanoreactor effectively promotes the formation of surface-adsorbed ˙CH and ˙CH$_2$ species, as well as gas-phase CH$_2$ radicals, leading to enhanced yields of C$_2$H$_2$ and C$_2$H$_4$.

## Density functional theory (DFT) calculations

Synchrotron radiation-based X-ray absorption spectroscopy (XAS) was employed to elucidate the chemical state and local structure of the WMO/m-SiO$_2$ catalysts. The X-ray absorption near-edge structure (XANES) spectra at the Mn K-edge (Mn$_3$O$_4$/m-SiO$_2$) and W K-edge (Na$_2$WO$_4$/m-SiO$_2$) in WMO/m-SiO$_2$ closely resembled those of the

individual Mn$_3$O$_4$/m-SiO$_2$ and Na$_2$WO$_4$/m-SiO$_2$ reference catalysts (Supplementary Fig. 34). This observation, in substantial contrast to the combined Na$_2$WO$_4$-Mn$_3$O$_4$ spectrum, strongly suggests the presence of isolated Mn$_3$O$_4$ and Na$_2$WO$_4$ active sites dispersed on the m-SiO$_2$ support. Further validation was provided by Fourier transform (FT) extended X-ray absorption fine structure spectroscopy (EXAFS) results (Fig. 4a, b). The peaks at 1.4 Å (Fig. 4a) and 1.2 Å (Fig. 4b) mainly represent the single scattering of Mn−O and W−O bonds, respectively, confirming the existence of isolated Mn and W species and the absence of direct Mn-W binding in the WMO/m-SiO$_2$ catalyst. The bond lengths for Mn−O and W−O in WMO/m-SiO$_2$ were determined to be 1.86 and 1.69 Å, respectively (Supplementary Table 3). The proposed Mn-O model exhibited excellent agreement with the experimental spectra, as evidenced by its superior fit in the XANES analysis and negligible deviations from DFT calculations (Supplementary Figs. 34−36).

To elucidate the fundamental mechanism of C$_2$H$_2$ and C$_2$H$_4$ production within WMO/m-SiO$_2$, spin-polarized periodic DFT calculations were employed to unravel the distinct roles of its components during plasma-catalytic NOCM reaction. Based on experimental results

(Fig. 1e and Supplementary Fig. 3), we selected $Mn_3O_4$ (211) and $Na_2WO_4$ (111) surfaces as model systems (Supplementary Fig. 35). The calculated binding energies of reaction intermediates involved in $CH_4$ dehydrogenation and subsequent coupling to $C_2H_2$ and $C_2H_4$ are presented in Supplementary Tables 4, 5. Notably, the binding strength of the $C_xH_y$ intermediates followed the order of $Mn_3O_4$ (211) > $Na_2WO_4$ (111), with intermediates binding via C or C-C on both surfaces (Supplementary Figs. 37, 38 and Supplementary Tables S4, S5).

The calculated Gibbs free energy change profiles for $CH_4$ dehydrogenation and intermediate active species coupling on the $Mn_3O_4$ (211) and $Na_2WO_4$ (111) catalysts are shown in Fig. 4c−e. For $CH_4$ dehydrogenation (Fig. 4c), the $Na_2WO_4$ (111) surface can facilitate $CH_4$ dehydrogenation to produce $^*CH_3$, $^*CH_2$, and $^*CH$ species. The most challenging step was $^*CH_3$ dehydrogenation, with an uphill energy of 0.22 eV, followed by $^*CH_2$ dehydrogenation, with an uphill energy of 0.19 eV on the $Na_2WO_4$ (111) surface. In contrast, $CH_4$ dehydrogenation on the $Mn_3O_4$ (211) surface required significantly higher uphill energy (0.73 and 0.75 eV) for producing $^*CH_2$ and $^*CH$ species, respectively. In addition, the reaction energy barriers for $^*CH_3 \rightarrow {}^*CH_2 + {}^*H$ and $^*CH_2 \rightarrow {}^*CH + {}^*H$ were calculated (Supplementary Table 8). These two elementary reactions were identified as the rate-determining steps in $CH_4$ dehydrogenation to generate $^*CH_3$, $^*CH_2$, and $^*CH$ species on both surfaces (Fig. 4c), with energy barriers of 0.97 and 0.69 eV on $Na_2WO_4$ (111), significantly lower than those on $Mn_3O_4$ (211) (1.93 and 2.02 eV). This indicates that the $Na_2WO_4$(111) surface is more effective in facilitating $CH_4$ dehydrogenation, consistent with experimental results (Supplementary Fig. 21). The $Mn_3O_4$ (211) surface was found to be more favorable for coupling $^*CH_2$ and $^*CH$ species to generate $C_2H_4$ and $C_2H_2$ (Fig. 4d, e). The most challenging steps were the desorption of $^*C_2H_4$ (Fig. 4d) and $^*CH_2$ dehydrogenation (Fig. 4e), with uphill energies of 0.49 and 0.19 eV, respectively. In contrast, on the $Na_2WO_4$ (111) surface, the most difficult steps were $^*CH_3$ dehydrogenation (Fig. 4d) and desorption of $^*C_2H_2$ (Fig. 4e), with significantly higher uphill energies of 0.73 and 2.52 eV, respectively. In addition, we investigated $CH_4$ dehydrogenation and radical coupling reactions under plasma-only conditions (Fig. 4f). The results suggest that plasma-driven NOCM without a catalyst favors the production of $C_2H_6$, in stack contrast to plasma-catalyzed reactions.

To elucidate the role of $SiO_2$ in the WMO/m-$SiO_2$ nanoreactor, $SiO_2/Mn_3O_4$ (211) and $SiO_2/Na_2WO_4$ (111) models were built (Supplementary Figs. 36, 39, 40). The calculated binding energies of reaction intermediates involved in $CH_4$ dehydrogenation and subsequent coupling to form $C_2H_2$ and $C_2H_4$ are presented in Supplementary Tables 6 and 7, with the most stable adsorption configurations illustrated in Supplementary Figs. 39, 40. Notably, $SiO_2/Na_2WO_4$ (111) exhibited significantly weaker binding for $^*C_2H_2$ compared to the isolated $Na_2WO_4$ (111) surface, with adsorption energies of −0.54 versus −0.95 eV, respectively (Supplementary Fig. 41 and Supplementary Tables 6, 7). In contrast, $SiO_2/Mn_3O_4$ (211) favored the desorption of $^*C_2H_4$ compared to pure $Mn_3O_4$ (211), with an adsorption energy of −0.62 eV (compared to −1.58 eV on pure $Na_2WO_4$ (111)). The binding energies of $^*CH/^*CH_2$ on $SiO_2/Mn_3O_4$ (211) and $SiO_2/Na_2WO_4$ (111) were −6.75 and −5.77 eV and −2.61 and −5.43 eV, respectively, indicating that $SiO_2/Mn_3O_4$ has a stronger adsorption capacity for $^*CH_2$ and $^*CH$ than $SiO_2/Na_2WO_4$. In summary, DFT calculations reveal that $Na_2WO_4$ facilitates $CH_4$ dehydrogenation to $^*CH$ or $^*CH_2$ intermediates, while $Mn_3O_4$ promotes the coupling of $^*CH$ and $^*CH_2$ to form $C_2H_2$ and $C_2H_4$, respectively. Furthermore, the presence of $SiO_2$ in combination with $Na_2WO_4$ and $Mn_3O_4$ enhances the desorption of the generated $^*C_2H_2$ and $^*C_2H_4$ species. These findings are consistent with experimental observations.

### Reaction mechanisms
The enhanced catalytic performance of $Na_2WO_4$-$Mn_3O_4$/m-$SiO_2$, compared to $Na_2WO_4$/m-$SiO_2$ and $Mn_3O_4$/m-$SiO_2$, highlights the synergistic effects between $Na_2WO_4$ and $Mn_3O_4$ sites (Supplementary Fig. 21). Quasi-in situ DRIFTS characterization (Fig. 3e and Supplementary Figs. 31, 32) indicates that W sites promote $^*CH_2$ generation, while Mn sites facilitate the coupling of $^*CH_2$ to form C=C bonds.

If radicals were fully converted at $Na_2WO_4$ before reaching $Mn_3O_4$ sites within m-$SiO_2$, significant carbon deposition would be expected on $Na_2WO_4$. However, compared to $Na_2WO_4$/m-$SiO_2$, the 1% $Na_2WO_4$-5% $Mn_3O_4$/m-$SiO_2$ catalyst exhibits lower carbon accumulation (Supplementary Fig. 21) and higher $^*CH_2$ and C=C intensities (Fig. 3e). These findings suggest that radicals initially generated on $Na_2WO_4$ sites undergo further transformation into $^*CH$ and $^*CH_2$, which subsequently migrate to $Mn_3O_4$ sites for coupling reactions to produce $C_2H_2$ and $C_2H_4$.

Wavelet transforms of the EXAFS spectra at the Mn K-edge and W K-edge for WMO/m-$SiO_2$, 5% $Mn_3O_4$/m-$SiO_2$, and 1% $Na_2WO_4$/m-$SiO_2$ reveal similar Mn−O and W−O scattering peaks at (4.1, 1.4 Å) and (3.9, 1.2 Å), respectively (Fig. 4g). These findings indicate that $Na_2WO_4$ and $Mn_3O_4$ are independently distributed on the m-$SiO_2$ support, consistent with TEM observations (Supplementary Fig. 3). Notably, TEM analysis also demonstrates the close spatial proximity of $Na_2WO_4$ and $Mn_3O_4$, providing potential pathways for intermediate species migration.

DFT calculations further reveal that the adsorption energies of $^*CH$ and $^*CH_2$ on $Mn_3O_4$ are −5.21 and −3.96 eV, respectively, significantly stronger than those on $Na_2WO_4$ (−3.27 and −3.08 eV) (Supplementary Tables 6, 7). Notably, when supported on m-$SiO_2$, the adsorption energy gap increases, suggesting that radicals preferentially stabilize on $Mn_3O_4$ rather than remaining on $Na_2WO_4$ (Supplementary Fig. 41). This thermodynamic preference, combined with the close proximity of two active sites, indicates that $^*CH$ and $^*CH_2$ species likely undergo surface diffusion or desorption-reabsorption migration, facilitating C-C coupling on $Mn_3O_4$. Similar bifunctional catalysis mechanisms have been reported in thermal catalysis systems, where intermediate spillover between distinct active sites enhances reaction efficiency[37,38].

The distribution and synergy between $Na_2WO_4$ and $Mn_3O_4$ sites further facilitate the sequential activation of C-H bonds and C-C coupling within the WMO/m-$SiO_2$ nanoreactor. Herein, a tandem reaction mechanism is proposed for $CH_4$ conversion to $C_2H_2$ and $C_2H_4$ (Fig. 3h): (I) $CH_4$ dissociation and $CH_3$ diffusion: Energetic electrons from the plasma induce $CH_4$ dissociation into $CH_x$ fragments, which subsequently diffuse toward the interior of the silica sphere due to the concentration gradient across m-$SiO_2$. (II) Dehydrogenation at $Na_2WO_4$ sites (support on the channel): $^*CH_3$ species undergo dehydrogenation at W sites, forming surface-adsorbed $^*CH$ and $^*CH_2$ species. (III) Surface species coupling on Mn sites: $^*CH$ and $^*CH_2$ species migrate from $Na_2WO_4$ to $Mn_3O_4$ sites, leading to C-C coupling for $C_2H_2$ and $C_2H_4$ production.

Notably, the mesoporous channels of m-$SiO_2$ restrict plasma penetration into the interior of m-$SiO_2$, thereby mitigating excessive $CH_4$ activation (e.g., methane cracking) and suppressing carbon deposition. This structural confinement, combined with the synergistic tandem catalysis of $Na_2WO_4$ and $Mn_3O_4$, significantly enhances the yield of $C_2$ products. Moreover, the shielding effect of the nanoreactor reduces plasma-induced product decomposition and recombination, further enhancing selectivity.

## Discussion
This work presents a promising nanoreactor catalyst design strategy that significantly improves the yield and selectivity of $C_2H_2$ and $C_2H_4$ through plasma-catalytic NOCM under mild conditions. The nanoreactor features a hollow nanosphere structure with $Na_2WO_4$ nanoparticles anchored on the interconnected channels and monodispersed $Mn_3O_4$ nanocrystals hosted within the internal cavity. By positioning the nanoreactors in the discharge area, methane conversion reached

34%, with a selectivity of 42.3% toward $C_2H_2$ and $C_2H_4$. This represents a nearly 4.5-fold increase in yield and a fourfold increase in selectivity for unsaturated $C_2$ hydrocarbons compared to the plasma-only system. Importantly, no deactivation was observed during the 25-h catalyst stability test. Mechanistic investigations revealed that $Na_2WO_4$ promotes the dehydrogenation of diffused $CH_4$ and $CH_3$, leading to the formation of $^{\cdot}CH$ and $^{\cdot}CH_2$ intermediates. These species subsequently undergo C-C coupling on the $Mn_3O_4$ surface to form $C_2H_2$ and $C_2H_4$. The excellent catalytic performance, supported by in situ plasma-coupled FTIR characterization, is further corroborated by DFT calculations. These calculations demonstrate that a tandem catalytic effect is achieved through the isolated $Na_2WO_4$ and $Mn_3O_4$ active sites, which are responsible for the enhanced selectivity. Furthermore, the mesoporous nanoreactor design prevents the reduction of internal $Mn_3O_4$ by $CH_4$ plasma through the Debye shielding effect. This reduces carbon deposition on the catalyst and protects the generated $^{\cdot}CH$ and $^{\cdot}CH_2$ intermediates from further decomposition due to the absence of plasma discharge within the mesopores. This catalyst design strategy offers significant potential for advancing plasma-catalysis. It enables highly selective and directional conversion, improving the energy efficiency of plasma-catalytic systems and paving the way for a high-value route to transform methane into unsaturated light olefins under mild conditions.

## Methods

### Synthesis of $SiO_2$ nanospheres
Mesoporous $SiO_2$ (m-$SiO_2$) was synthesized using a double template method in an ethanol solution. First, ethanol and polyacrylic acid were added to a vial and stirred for 30 min. Then, diluted ammonia water was added to the above solution, followed by the introduction of polyether, and the mixture was stirred for another 30 min. After that, ethyl orthosilicate was added dropwise to the vial, resulting in a suspension after 4 h of stirring. Subsequently, the suspension was centrifuged and washed three times with ethanol. Finally, the precursors were evaporated overnight and calcination at 550 °C.

### Synthesis of α $Na_2WO_4$-β $Mn_3O_4$/m-$SiO_2$
α $Na_2WO_4$/m-$SiO_2$, β $Mn_3O_4$/m-$SiO_2$, and α $Na_2WO_4$-β $Mn_3O_4$/m-$SiO_2$ (where α and β denote the respective weight percentages of each metal oxide) were synthesized using the incipient wetness method. The Mn loading on the m-$SiO_2$ support varied from 2 to 10 wt% (2, 5, 7.5, and 10 wt%), while the $Na_2WO_4$ loading ranged from 0.5 to 5% (0.5, 1, 2, and 5 wt%). For the preparation of $Na_2WO_4$/m-$SiO_2$ and $Mn_3O_4$/m-$SiO_2$, aqueous solutions of manganese nitrate tetrahydrate ($Mn(NO_3)_2 \cdot 4H_2O$) or sodium tungstate dihydrate ($Na_2WO_4 \cdot 2H_2O$) were mixed with m-$SiO_2$ in a water bath. The mixture was stirred until it reached a paste-like consistency and then dried overnight at 70 °C. Subsequently, the dried samples were calcined in air at 550 °C. A two-step impregnation method was used for the synthesis of β $Na_2WO_4$-α $Mn_3O_4$/m-$SiO_2$ (denoted as WMO/m-$SiO_2$). First, $Mn_3O_4$/m-$SiO_2$ was prepared as described above. Then, $Na_2WO_4$ was loaded onto the $Mn_3O_4$/m-$SiO_2$ using the same drying and calcination procedures. This method was also used to prepare 1% $Na_2WO_4$-5% $Mn_3O_4$/$SiO_2$ and 1% $Na_2WO_4$-5% $Mn_3O_4$/ZSM-5. Before testing, all catalysts were crushed and sieved to obtain particles with sizes between 30 and 60 mesh.

### Synthesis of $Mn_3O_4$-deposited m-$SiO_2$ catalysts
Three types of catalysts with varying $Mn_3O_4$ locations were prepared: In-m-$SiO_2$, where $Mn_3O_4$ was deposited exclusively within the mesopores of m-$SiO_2$; Both-m-$SiO_2$, where $Mn_3O_4$ was partially distributed within the mesopores and on the exterior surface of m-$SiO_2$; Out-m-$SiO_2$, where $Mn_3O_4$ was primarily located on the exterior surface of m-$SiO_2$. In-m-$SiO_2$ was synthesized as described above. Both-m-$SiO_2$ was prepared using a rapid heating and drying method, where $Na_2WO_4$

was first loaded onto m-$SiO_2$, followed by the deposition of $Mn_3O_4$. The precursors were rapidly heated from room temperature to 300 °C at a heating rate of 20 °C min$^{-1}$ for 2 h and then calcined at 550 °C. Out-m-$SiO_2$ was synthesized by mechanically mixing pre-synthesized Mn-O and Na-W-O precursors. Specifically, solutions of Mn or W were dissolved in deionized water and stirred at 50 °C for 3 h. After drying overnight, the precursors were mechanically mixed with m-$SiO_2$ and calcined at 550 °C.

### Catalyst activity test
The performance of the catalyst in the thermal catalytic NOCM reaction (denoted as catalyst only) was evaluated in a DBD reactor (plasma off) equipped with heating tape and operated at atmospheric pressure. The reactor was loaded with 1% $Na_2WO_4$-5% $Mn_3O_4$/m-$SiO_2$ and secured with quartz wool on both ends. Prior to testing, the catalyst was pretreated with argon (200 mL min$^{-1}$) at 100 °C for 20 min to remove impurities. The temperature was then increased to 250 °C and monitored using a K-type thermocouple placed within the catalyst bed. A diluted methane feed (5 vol% $CH_4$ in Ar) at a total flow rate of 200 mL min$^{-1}$ was used to minimize mass transfer limitations and accurately evaluate the intrinsic catalytic activity. The feed gas was preheated to 30 °C, and the experiments were conducted for 60 min.

For plasma-catalysis and plasma-only conditions, the DBD reactor was operated without external heating. The experimental procedure involved the following steps: (1) The catalyst was pretreated with argon (100 mL min$^{-1}$) at 100 °C for 20 min, followed by cooling to room temperature. (2) A 5 vol% $CH_4$/Ar mixture (200 mL min$^{-1}$) flowed through the DBD reactor for 10 min before switching on the plasma. (3) The plasma was ignited at an SEI of 5.1 kJ L$^{-1}$ (calculated as the discharge power divided by the gas flow rate) and maintained at a discharge power of 17 W with a flow rate of 200 mL min$^{-1}$ for 60 min. (4) After switching off the plasma, the DBD reactor was purged with argon (100 mL min$^{-1}$) for 10 min. (5) The spent catalyst was removed, and a 10 vol% $O_2$/Ar mixture (100 mL min$^{-1}$) was introduced to oxidize solid carbon deposited in the DBD reactor. This step was carried out at an SEI of 6.14 kJ L$^{-1}$ until no CO or $CO_2$ was detected in the exhaust gas. (6) The spent catalyst was reintroduced into the cleaned reactor, and a 10 vol% $O_2$/Ar mixture (100 mL min$^{-1}$) was used to oxidize carbon deposited on the catalyst. This step was continued, and no CO or $CO_2$ were detected in the exhaust gas. The amount of carbon deposited was calculated using the equation $N_C = V \times (C_{CO2} + C_{CO}) \times 22.4$, where $V$ is the total volume of exhaust gas during the oxidation process, and $C_{CO2}$ and $C_{CO}$ are the concentrations of $CO_2$ and CO in the exhaust gas, respectively. To avoid interfering with the plasma field, the reactor temperature was measured using an infrared thermometer.

### Isotopic labeling experiments
To investigate the relative contributions of $CH_2$ and $CH_3$ radicals to $CH_4$ decomposition, isotopic labeling experiments were conducted. The catalyst was first pretreated in a 5 vol% $CH_4$/Ar mixture (discharge power 17 W, flow rate 200 mL min$^{-1}$) for 60 min and then cooled to room temperature. Next, argon (200 mL min$^{-1}$) was flowed through the reactor for 10 min to purge residual gases. A gas mixture of 2.5 vol% $CD_4$ and 2.5 vol% $CH_4$ in argon (200 mL min$^{-1}$) was subsequently introduced into the DBD reactor for 10 min. The plasma was then switched on and sustained for 60 min, during which mass spectrometric signals at m/z = 15, 18, and 19 were monitored. The concentrations of $CH_4$ and $CD_4$ in the mixture were determined using mass spectrometry. From these measurements, the conversion of $CH_4$ and $CD_4$, as well as the selectivity of the product, was determined. Under plasma discharge conditions, $CH_4$ and $CD_4$ dissociate into radicals including $CH_3$, $CH_2$, CH, $CD_3$, $CD_2$, CD, H, and D. These radicals recombine to form isotopically labeled methane species, such as $CD_3H$

(m/z = 19), and $CD_2H_2$ (m/z = 18). The formation of $CD_3H$ and $CD_2H_2$ results from the recombination reactions of $CD_3$ with H and $CD_2$ with two H atoms, respectively. The signals at m/z = 15 and m/z = 19 reflect the concentrations of $CH_4$ and $CD_3H$, respectively. When analyzing the peak intensity of $CD_2H_2$, it is critical to consider the contribution of $CD_4$ fragmentation (m/z = 20), which generates $CD_3$ (m/z = 18). Specifically, the signal at m/z = 18 reflects contributions from both $CD_2H_2$ and the $CD_3$ fragment derived from $CD_4$ fragmentation.

### In situ plasma-coupled FTIR characterization

To elucidate plasma-induced surface reactions during the plasma-catalytic NOCM process, we employed in situ plasma-coupled FTIR spectroscopy using a custom-designed plasma gas cell[19]. The experimental procedure is described in detail as follows: (I) Prior to analysis, each sample was pretreated with argon plasma (99.999% purity) at a flow rate of 100 mL min⁻¹ in a DBD reactor at 100 °C for 20 min to remove residual surface species. (II) A 5% $CH_4$/Ar mixture (40 mL min⁻¹) was introduced to purge the cell for 30 min. During this step, the temperature was decreased from 100 to 35 °C, after which the IR background spectrum was collected. (III) The plasma was switched on, and the plasma-catalytic NOCM was conducted for 15 min. (IV) After switching off the gas flow, IR spectra were collected every 3 min for a total duration of 18 min.

Quasi-in situ diffuse reflectance infrared Fourier transform spectroscopy (DRIFT) analysis was conducted using an FTIR spectrometer (IS50, Thermo Fisher Co. Ltd.) equipped with a liquid nitrogen $N_2$-cooled mercury-cadmium-telluride (MCT) detector. The background spectrum was obtained by pretreating the catalyst with Ar plasma in a DBD reactor under conditions identical to those used in the plasma-catalytic NOCM experiments (SEI = 5.1 kJ L⁻¹, flow rate = 200 mL min⁻¹). Following pretreatment, the catalyst was cooled to room temperature before conducting the quasi-in situ DRIFTS measurements. To avoid air exposure, the pretreated catalysts were transferred from the DBD reactor to the DRIFT cell within a glovebox. For the plasma-catalytic NOCM reaction, the plasma was switched off after the experiment, and the inlet and outlet of the DBD reactor were sealed. Subsequently, the spent catalysts were then transferred from the DBD reactor to the DRIFT cell within a glovebox, and IR spectra were collected at room temperature.

### Computational details

Spin-polarized density functional theory (DFT)[39,40] calculations were performed using the Vienna ab initio simulation package (VASP) code[41]. The exchange-correlation interactions between electrons were described using the Perdew–Burke–Ernzerhof (PBE) functional within the generalized gradient approximation (GGA)[42,43]. Following convergence tests, plane-wave pseudopotentials with kinetic cutoff energy of 420 eV[44] for $Mn_3O_4$ and 500 eV[45] for $Na_2WO_4$ method were employed within the projector augmented wave (PAW) method. The $Mn_3O_4$ (211) and $Na_2WO_4$ (111) surfaces were selected as the computational models based on the experimental results (Fig. 1e and Supplementary Fig. 3). To minimize interactions between the slab and its periodic images, a vacuum layer of ~15 Å was added above the slab. During geometry optimization, the bottom two atomic layers were fixed, while all other atoms and adsorbates were allowed to relax until the force on each atom was less than 0.01 eV Å⁻¹. A convergence criterion of $1 \times 10^{-5}$ eV/atom was used for structural optimization. Brillouin zone integration was performed using a $2 \times 2 \times 1$ Monkhorst-Pack grid with a Methfessel-Paxton smearing width (σ) of 0.2 eV. Due to the presence of localized 3$d$ states on Mn, the electronic structure of Mn was treated within the DFT + U formalism with a U-J parameter of 4.00 eV[46]. In addition, to account for weak interactions within the catalyst, van der Waals corrections were incorporated using the DFT-PBE-D3 method[47]. Further details regarding the DFT calculation methods are provided in the Supplemental Information.

## Data availability

The data presented in the figures and the key findings of this study are available from the corresponding authors upon reasonable request. Source data are provided with this paper.

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

## Acknowledgements

This work was financially supported by the National Natural Science Foundation of China (Nos. U2202251 and 52174279), the Applied Basic Research Program of Yunnan Province for Distinguished Young Scholars (No. 202201AV070004), the Key Project (No. 202401AS070062), and the Yunnan Provincial Science and Technology Project at Southwest United Graduate School (No. 202302AO370017). X.T. acknowledges the funding from the European Union's Horizon Europe research and innovation program under Grant Agreement No. 101069931 and the UKRI Horizon Europe Guarantee Fund (Ref. 10038857).

## Author contributions

C.L., K.L., X.T. and H.W. conceived the idea and designed the experiments. C.L. and R.X. synthesized and evaluated the catalysts, performing catalyst characterization and analysis. Y.W. and C.L. performed plasma diagnostics and in situ plasma-coupled FTIR characterization and analysis. D.T. performed the DFT calculations. R.W., S.X. and W.L. conducted the XAFS characterization and analysis. C.L., K.L. and X.T. wrote the paper. All authors discussed the results and commented on the manuscript.

## Competing interests

The authors declare no competing interests.
