## [Transparent Peer Review File · Nature Communications]

Shielded Bifunctional Nanoreactor Enabled Tandem Catalysis for Plasma Methane Coupling

Corresponding Author: Professor Xin Tu

This file contains all reviewer reports in order by version, followed by all author rebuttals in order by version

Version 0:

Reviewer comments:

Reviewer #1

(Remarks to the Author)
Summary

In this article, the authors propose a novel shielded bifunctional nanoreactor for the plasma-assisted catalysis of methane conversion to unsaturated C₂ hydrocarbons. They show that by using mesoporous silica nanoparticles in conjunction with Na₂WO₄ and Mn₃O₄ catalytic site, the selectivity and production can be improved. They argue this is due to the shielding of the catalytic Na₂WO₄ and Mn₃O₄ nanoparticles from the plasma by the narrow channels of the mesoporous SiO₂. They motivate these claims with a plethora of experimental techniques and density functional theory calculations. The experimental measurements largely provide convincing evidence of the suggested reaction mechanism. The DFT calculations are less well-founded, having made debatable approximations. I recommend publishing after revision.

Major comments

1. I find the motivation for the chosen surface facet lacking, even if it is the most prevalent facet (needs proof or reference), this is no guarantee that this is also the facet that hosts the active sites. More literature is needed to motivate the choice of surface facet and justify its relevance.
2. On a similar note, the additions of the SiO₂ on the surface is never well-defined in the computational methods section, nor is it clear that the approach taken is in any way representative of the system. For example, looking at Fig S34, it seems to me as if the oxide surfaces are the catalyst support, and that the SiO₂ is just a small cluster (basically Si₂O₄) adsorbed on that surface. However, in the experiments, SiO₂ is the support, and the oxides are supposedly much smaller nanoparticles supported by SiO₂. This needs more clarifications and justification in order to be eligible for the conclusion.
3. Furthermore, in the way the DFT calculations are reported now, they are not reproducible. For example the version of VASP is missing and there is no specifications of the used pseudopotentials. Additionally, the specified thickness of the vacuum layer as "approximately 15 Å" is needlessly vague for these calculations. Also connected to the previous point, an archive with the final geometries of the various calculations will considerably improve the reproducibility.
4. In general, the figures in the main paper of low quality. They do not fulfill the pixel density, nor the minimal font size specified by the journal. This makes figures hard to read and interpret. The figures in the supporting information, especially S7, S8, S9, S29, are even worse sometimes being completely illegible.
5. Langmuir-Rideal (LR; see below) reactions are reported here to play an important role ("Na₂WO₄ may facilitate the dehydrogenation of CH_x radicals to *CH and *CH₂ species⁹² on the surface via the Eley-Rideal (E-R) mechanism."). However, several atomistic dynamical simulations have reported that LR reaction rates might be overestimated compared to hot atom-mediated reaction, where a reactant would preferentially first adsorb on the surface, and not equilibrate (sufficiently) with the surface before another adsorbate is encountered and reaction occurs.
6. In the proposed reaction mechanism the authors state that the *CH and *CH₂ migrate from the Na₂WO₄ to the Mn sites, which is not supported by the results from the article. Consider specifying the nature of the migration: via desorption or via surface diffusion and providing supporting references to back up this claim. Especially since the authors also claim that

these active sites are distributed independently, i.e., they are isolated.

Minor Comments

Next follow some minor comments:

1. The authors refer to the Eley-Rideal mechanism for surface reactions. I would prefer the name Langmuir-Rideal instead of Eley-Rideal as motivated by "Prins, R. (2018-06-01). "Eley–Rideal, the Other Mechanism". *Topics in Catalysis*. 61 (9): 714–721. doi:10.1007/s11244-018-0948-8. ISSN 1572-9028". A brief summary of the article is that Dan Eley and Eric Rideal never worked on the supposed Eley-Rideal mechanism of gas phase molecule reacting with an adsorbate, which was already proposed by Langmuir. Instead, they worked on a similar but distinct reaction mechanism between a physisorbed and a chemisorbed molecule.
2. In the supplementary information, figures 30 and 31 are mislabeled as 27 and 28.
3. The term DBD is sometimes used to refer to the device instead of the actual discharge, use DBD reactor instead. (line 226, 292)
4. Many acronyms are never expanded or not at first occurrence: XRD, SEM, ZSM-5, TPR, XPS, DRIFTS, (NTP-)CM.
5. units on line 146 and 153 are different, making comparison difficult.
6. The authors state that C₂H₂ and C₂H₄ fractions are quadrupled for WMO/m-SiO₂, while figure 1c does not support this. There one sees a change from ~20% to ~40%, which would suggest a doubling instead of quadrupling.
7. Figure 3h references on line 380 does not exist.
8. In the conclusion (line 422), the author mentions the potential of combinations with technical advances to produce competitive reactor design, but mentions no examples nor references any works. Without those, this statement is too general and vague and does not support the narrative.
9. Is the added energy in the thermal catalysis case similar to the plasma + catalyst without heating case? In other words, what is the efficiency of the various ways of adding energy? It would be a further indication of the efficiency of the plasma-assisted catalytic process compared to the thermal catalytic process.
10. Only adsorption energies are calculated, not barriers. A comment in the paper about the approximation made here (barrier height scales with adsorption energies) should be mentioned. If literature results for a (few) barrier(s) in this work exist, they should be discussed.
11. The DFT method section in the main paper should mention PBE-D3 (like the SI), instead of only PBE (it has a dramatic influence on the results).

Reviewer #2

(Remarks to the Author)

Reviewer #3

(Remarks to the Author)

This is a thorough work on the non-oxidative coupling of methane via plasma-catalysis. The main novelty of the work lies on the design of catalytic materials, wherein mesoporous SiO₂ particles with hollow structures are loaded with Na₂WO₄ within the SiO₂ channels and Mn₃O₄ in the cavities of SiO₂. A range of experimental, characterisation and computational methods are used to obtain insights on the performance of these materials. It is discussed that Na₂WO₄ enhances the conversion of CH₄ and CH₃, Mn₃O₄ promotes the coupling of surface CH and CH₂ to C₂H₂ and C₂H₄, and SiO₂ prohibits the plasma from reaching in the cavity. The work is very interesting and novel in that it presents a catalyst design approach targeted at a plasma-catalysis system, in contrast to most works in the field that follow design approaches typical for conventional heterogeneous catalysis. Nonetheless, below a range of comments are presented that should be addressed by the authors in a revised version before consideration for publication can be made.

In all figures of the manuscript the selectivity of C₂H₂ and C₂H₄ is presented as a lump. Considering that the main design target of the presented materials is the enhancement of the yield/selectivity of C₂ species other than ethane, it is not clear why the manuscript omits details on this regard. Given the GC and MS experimental details provided, the authors clearly can separate the two species C₂H₂ and C₂H₄ in the reaction products and provide detailed selectivities. Supplementary Fig. 21b is the only figure in the entire manuscript that does show separate traces for C₂H₂ and C₂H₄. It would be necessary to

update all figures in the main text and supplementary information to show the selectivities for C₂H₂ and C₂H₄ separately. There is a substantial difference if the presented lumped yields have been achieved through a high selectivity in C₂H₂ versus C₂H₄. There are numerous plasma-based methods already presented in literature that have achieved very high selectivity to C₂H₂, with many of these methods not even requiring a catalyst, for example those based on nanosecond pulsed discharges. On the contrary, achieving a high selectivity to C₂H₄ has to date been a major challenge. The discussion would need to be updated to reflect the above, depending on whether it is C₂H₂ or C₂H₄ that is primarily formed. In relation to the above comment, in lines 207 onwards the authors state that "... the yield of ethylene decreases from 7.7% (WMO/m-SiO₂) to 4.7% (Na₂WO₄-Mn₃O₄, denoted as WMO)" referring to Fig. 2b and Supplementary Fig. 19. Firstly, the authors likely mean Fig. 2d and not Fig. 2b. In all cases, both in Fig. 2d and Supplementary Fig. 19, where these results are presented, only lumps of ethylene and acetylene are shown.

The manuscript does not determine anywhere what was the power applied, as determined by the Lissajous plots shown in Supplementary figures 11-12. There is only a mention of a specific energy input (SEI) value in the Methods section. What was the power and flowrates that resulted in this SEI? Were these parameters constant for all experiments carried out? There are a range of different cases studied in this work, so for clarity it would be more appropriate if all figure captions explicitly stated the applied power, flowrate and SEI.

In continuation to the above comment, and more importantly, there is no discussion at all in the manuscript on what is the energy efficiency or energy cost of the process. The obtained results need to be compared on the one hand with those from literature not only in terms of yield achieved, but also in terms of the energy cost for the conversion of CH₄ and the production of C₂H₄. Additionally, the energy efficiency of the process needs to be presented and discussed, comparing specifically the energy cost of the plasma generation to the reaction enthalpy of the overall methane conversion.

The feed stream comprised of a heavily diluted mixture at a 5% CH₄ in Ar, however the role of Ar is not discussed in the manuscript. There are multiple experimental works (e.g. those of Jo et al. 10.1063/1.4818795 and 10.1016/j.ces.2015.03.019) and computational studies (e.g. that of Maitre et al. 10.1016/j.ces.2022.117731) that extensively discuss the major contribution of Ar in the activation of methane though a range of energy transfer mechanisms, including Penning ionisations and dissociations. The yields achieved are very likely affected by the presence of Argon in the reactant mixture and would be much lower in a pure CH₄ feed. Similarly, carbon deposition would be significantly different and likely more pronounced with a pure CH₄ stream. Appropriate consideration to this needs to be given in the discussion and ideally results from equivalent experiments with pure CH₄ need to be presented.

Supplementary figure 12 depicting electrical signals shows that the current has a peculiar triangular shape. Normally, a sinusoidal signal with superimposed spikes due to microdischarges is expected. What is the reason for this signal profile? The manuscript Supplementary information should provide details on how the current signal was measured. For completeness and for comparison, the Supplementary information should also show the time resolved V and Q signals, besides the Lissajous plots.

The authors stress the importance of balancing the oxygen content in the catalysts and also carry out comparative experiments with pre-reduced samples to demonstrate this. Considering that H₂ is a primary product of the coupling reaction, are the oxides expected to be eventually reduced? The plasma operation at extended periods will unavoidably lead to parasitic heating of the catalyst bed.

More details need to be provided on the isotopic labelling experiments, namely the conversion and selectivities achieved for each of the presented experiments.

The lifetime of the radicals while diffusing in the mesoporous SiO₂ needs to be estimated and considered in the discussion. Depending on the diffusion length, the radicals can possibly preferentially recombine before reaching the inside of the particles for catalytic reactions. Also related to this comment, the contribution of plasma phase is largely neglected in the discussion and should be considered.

There are errors in the labelling of supplementary figures 30-31.

Version 1:

Reviewer comments:

Reviewer #1

(Remarks to the Author)

Overall, the authors have done a considerable amount of additional analysis, explanations and writing to improve the paper. The work presents an interesting way of designing a plasma reactor, with a reasonable understanding of the underlying mechanisms (it should be noted that in plasma catalysis a lot of uncertainties and unknowns are present). I find the science in this work very interesting and I now recommend publication of this paper, if the comments related to the figures' clarity below are resolved.

Most figures in the SI are now good with the exception of, at least, Figs. S30, 37, 38, 39, 40, which still display noticeable pixelation. In the main article, all figures still display noticeable pixelation and compression artifacts. My advice is to have a careful look at the settings of the plotting software. Normally, you can configure the absolute figure size and resolution, which should avoid all problems currently present with the figures. Note that it seems to me that in some places the ratio of the (sub)figure also might have changed. Another suggestion for clarity of figure 1c: Please include an arrow pointing to the second y-axis similar as in figure 1h.

Reviewer #2

(Remarks to the Author)

Reviewer #3

(Remarks to the Author)

The manuscript has been satisfactorily revised according to this reviewer's comments and is suggested for publication in its current version.

Response Letter

We sincerely thank the reviewers for constructive and valuable comments on our manuscript (Manuscript Number: NCOMMS-24-61933-T). We have thoroughly considered all the comments and have made substantial revision to the manuscript accordingly. Below, we provide a point-to-point response to the reviewers' comments. The revised portions in the manuscript and supporting information have been marked in red for easy reference.

Reviewer #1 (Remarks to the Author):

In this article, the authors propose a novel shielded bifunctional nanoreactor for the plasma-assisted catalysis of methane conversion to unsaturated C₂ hydrocarbons. They show that by using mesoporous silica nanoparticles in conjunction with Na₂WO₄ and Mn₃O₄ catalytic sites, the selectivity and production can be improved. They argue this is due to the shielding of the catalytic Na₂WO₄ and Mn₃O₄ nanoparticles from the plasma by the narrow channels of the mesoporous SiO₂. They motivate these claims with a plethora of experimental techniques and density functional theory calculations. The experimental measurements largely provide convincing evidence of the suggested reaction mechanism. The DFT calculations are less well-founded, having made debatable approximations. I recommend publishing after revision.

1. I find the motivation for the chosen surface facet lacking, even if it is the most prevalent facet (needs proof or reference), this is no guarantee that this is also the facet that hosts the active sites. More literature is needed to motivate the choice of surface facet and justify its relevance.

Reply:

We appreciate the reviewer's insightful suggestions. Based on the XRD pattern (Supplementary Figs. 4) and TEM images in Figure 1(e) and Supplementary Figs. 3, the surface facets with the interplanar spacing of 0.38 nm and 0.25 nm can be observed,

which correspond to the Na_2WO_4 (111) and Mn_3O_4 (211) surface, respectively. Therefore, Mn_3O_4 (211) and Na_2WO_4 (111) facets were chosen as the surface facet in the DFT calculations (on pages 4 to 5). In addition, we also reviewed a lot of literature to support our results (ACS Catal. 2014, 4, 4106-4115, J. Phys. Chem. C 2013, 117, 6218-6224, Applied Catalysis O: Open 193 (2024) 206979). Meanwhile, The relevant modeling and calculation details are also detailed in the Supplementary information (on pages 4 to 5)

Original:

In the manuscript:

To elucidate the fundamental mechanism of C_2H_2 and C_2H_4 production within $\text{WMO}/\text{m-SiO}_2$, spin-polarized periodic DFT calculations were employed to unravel the individual roles of the components in the $\text{WMO}/\text{m-SiO}_2$ nanoreactor during the plasma-catalytic methane coupling reaction.

Supplementary information:

As shown in Supplementary Figs. 33 to 40 and Tables S4 and S5, different potential adsorption sites for reaction intermediates produced during the plasma catalytic conversion of CH_4 to C_2H_2 and C_2H_4 on the Mn_3O_4 (211) and Na_2WO_4 (111) surfaces were considered during the calculation of binding energies (BE).

Revised in the manuscript:

In the manuscript: (Page 19, lines 346-350)

To elucidate the fundamental mechanism of C_2H_2 and C_2H_4 production within $\text{WMO}/\text{m-SiO}_2$, spin-polarized periodic DFT calculations were employed to unravel the distinct roles of its components during plasma-catalytic NOCM reaction. **Based on experimental results (Figure 1e and Supplementary Figs. 3), we selected Mn_3O_4 (211) and Na_2WO_4 (111) surfaces as model systems (Supplementary Fig. 35).....**

Supplementary information: (Pages 5 to 6)

Spin-polarized density functional theory (DFT)^{2,3} calculations were performed using the Vienna ab initio simulation package (VASP, version 6.21) code⁴. The exchange-correlation interactions between electrons were described using the Perdew–Burke–Ernzerhof (PBE) functional within the generalized gradient approximation

(GGA)^{5,6}. Plane wave pseudopotentials with kinetic energy cutoffs of 420 eV⁷ for Mn₃O₄ and 500 eV⁸ for Na₂WO₄ were employed within the projector augmented wave (PAW) method, as determined by convergence tests. The Mn₃O₄ (211) and Na₂WO₄ (111) surfaces were selected as computational models based on experimental results (Fig. 1e and Supplementary Fig. 3). The DFT-calculated lattice constants for bulk Mn₃O₄ were $a = b = 5.841 \text{ \AA}$ and $c = 9.462 \text{ \AA}$, while for Na₂WO₄, $a = b = c = 9.157 \text{ \AA}$. These values were all in excellent agreement with previously reported values, $a = b = 5.840 \text{ \AA}$ and $c = 9.500 \text{ \AA}$ for Mn₃O₄⁹, and $a = b = c = 9.129 \text{ \AA}$ for Na₂WO₄^{9,10}. The Mn₃O₄ (211) (Supplementary Fig. 35 (a, b)) and Na₂WO₄ (111) (Supplementary Fig. 35 (c, d)) surfaces were modeled using a four-layer 2×2 surface slab. A vacuum layer with a thickness of ~15 Å was added to the slab to eliminate unphysical interactions between periodic images perpendicular to the surface. During geometry optimization, atoms in the bottom two layers were fixed, while all other atoms, including adsorbates, were allowed to relax until the force on each ion was less than 0.01 eV Å⁻¹. The convergence criterion for structure optimization was set to 1×10⁻⁵. Brillouin-zone integration was performed using a 2×2×1 Monkhorst-Pack grid with Methfessel-Paxton smearing ($\sigma = 0.2 \text{ eV}$). To account for on-site coulomb interactions, the electronic structure of Mn was treated using the DFT+U formalism, with a parameter U-J = 4.00 eV applied to the localized 3d states of Mn¹¹. In addition, van der Waals corrections were incorporated using the DFT-PBE-D3 method to accurately describe weak interactions with the catalyst¹².

As shown in Supplementary Fig. 35(a-d), Supplementary Tables 4 and 5, different potential adsorption sites of intermediates for the plasma-catalytic NOCM to C₂H₂ and C₂H₄ on Mn₃O₄ (211) and Na₂WO₄ (111) surfaces were considered during the calculation of binding energies (BE). The BE of each adsorbate was calculated as follow¹²⁻¹⁵:

2. On a similar note, the additions of the SiO₂ on the surface is never well-defined in the computational methods section, nor is it clear that the approach taken is in any way representative of the system. For example, looking at Fig S34, it seems to me as if the

oxide surfaces are the catalyst support, and that the SiO₂ is just a small cluster (basically Si₂O₄) adsorbed on that surface. However, in the experiments, SiO₂ is the support, and the oxides are supposedly much smaller nanoparticles supported by SiO₂. This needs more clarifications and justification in order to be eligible for the conclusion.

Reply:

We appreciate the reviewer's insightful suggestions. In this study, we constructed NaWO₄/SiO₂ and Mn₃O₄/SiO₂ models using an inverse modeling approach. Specifically, a small SiO₂ cluster (Si₈O₄) was deposited on four-layer 2×2 Mn₃O₄ (211) (Supplementary Fig. 36(a, b)) and Na₂WO₄ (111) (Supplementary Fig. 36(c, d)) surfaces to investigate the adsorption behaviors of different intermediates during the plasma-catalytic conversion of CH₄ to C₂H₂/C₂H₄. The inverse modeling approach is accepted for many catalysts, as reported in references (Science, 355, 2017, 1296-1299; Chem, 2020, 6, 2703-2716; Nature Communications, 2019, 10:1166; Chem, 2020, 6 (2), 419-430).

In addition, we constructed a conventional model to validate the accuracy of the inverse model. As shown in Figure R1, the conventional model features Mn₃O₄ and Na₂WO₄ clusters supported on amorphous SiO₂. To verify the inverse model's reliability, we calculated the adsorption energies of different intermediates (*CH, *CH₂, and *C₂H₂, *C₂H₄) on both Mn₃O₄/SiO₂ and Na₂WO₄/SiO₂ surfaces, as presented in Tables 1 and 2, respectively. Direct comparisons of adsorption energies for key intermediates (*CH, *CH₂, and *C₂H₂, *C₂H₄) between the conventional model and the inverse model were summarized in Table 3. It should be highlighted that the modeling of amorphous oxides is very difficult, and the inverse modeling approach can save a lot of computing resources for the catalysts with amorphous oxides as supports.

Figure R1. DFT-optimized structure, front view and top view: Various possible adsorption sites for CH_x ($x = 4, 3, 2, 1$) adsorbed on $\text{Na}_2\text{WO}_4/\text{SiO}_2$ and $\text{Mn}_3\text{O}_4/\text{SiO}_2$ surfaces are marked in figures (a) and (c), respectively; various possible adsorption sites for C_xH_y ($x = 2, y = 6, 4, 2$) adsorbed on $\text{Na}_2\text{WO}_4/\text{SiO}_2$ and $\text{Mn}_3\text{O}_4/\text{SiO}_2$ are marked in figures (b) and (d), respectively. Mn-slate blue, O-red, Na-purple, Si-yellow and W-dark blue, respectively.

Table S1. DFT calculated binding energies (BE) of adsorbates on the $\text{Mn}_3\text{O}_4/\text{SiO}_2$ surface on various possible adsorption sites.

Mn ₃ O ₄ -SiO ₂ surface				
Adsorbate	Bound site	Bound via	BE (eV)	Comments
CH	Mn-Mn Bridge	/	/	/
	Mn-O Bridge	/	/	/
	Mn Top	/	/	/
	O Top1	/	/	/
	O Top2	/	/	/
	O Top3	/	/	/

	Si Top	C	-8.74	/
CH ₂	Mn-Mn Bridge	/	/	/
	Mn-O Bridge	C	-5.91	Move to O Top2
	Mn Top	C	-1.94	/
	O Top1	/	/	/
	O Top2	/	/	/
	O Top3	C	-3.51	/
	Si Top	C	-4.12	/
C ₂ H ₄	Mn-Mn Bridge	C	-1.01	Move to Mn Top
	Mn-O Bridge	C	-1.04	Move to Mn Top
	Mn Top	C	-1.10	/
	O Top1	/	/	/
	O Top2	/	/	/
	O Top3	/	/	/
	Si Top	C	-0.67	/

Table S2. DFT calculated binding energies (BE) of adsorbates on the Na₂WO₄/SiO₂ surface on various possible adsorption sites.

Na ₂ WO ₄ -SiO ₂ surface				
Adsorbate	Bound site	Bound via	BE (eV)	Comments
CH	Mn-O Bridge	/	/	/
	Mn Top	/	/	/
	Na-O Bridge	/	/	/
	Na Top	/	/	/
	O Top1	/	/	/
	O Top2	C	-5.06	/
	Si Top	/	/	/
CH ₂	Mn-O Bridge	C	-3.14	Move to O Top1
	Mn Top	C	-3.24	/
	Na-O Bridge	/	/	/
	Na Top	/	/	/
	O Top1	/	/	/
	O Top2	/	/	/
	Si Top	/	/	/
C ₂ H ₂	Mn-O Bridge	C	-0.31	Move to Mn Top
	Mn Top	C	-0.26	/
	Na-O Bridge	C	-0.82	Move to Na Top
	Na Top	C	-0.82	/
	O Top1	/	/	/

	O Top2	/	/	/
	Si Top	/	/	/

Table S3. The summarized direct comparisons of adsorption energies for key intermediates (*CH, *CH₂, and *C₂H₂, *C₂H₄) between the conventional model and the inverse model.

Catalyst model	Adsorption energy (eV)			
	C ₂ H ₂	C ₂ H ₄	CH	CH ₂
Mn ₃ O ₄	/	-1.58	/	/
Na ₂ WO ₄	-0.95	/	/	/
SiO ₂ /Mn ₃ O ₄ (211)	/	-0.62	-6.75	-5.77
SiO ₂ /Na ₂ WO ₄ (211)	-0.54	/	-2.61	-5.43
Mn ₃ O ₄ /SiO ₂	/	-1.10	-8.75	-5.91
Na ₂ WO ₄ /SiO ₂	-0.82	/	-5.06	-3.24

For the inverse models (see Supplementary Figs. 36, 39 and 40), the SiO₂/Na₂WO₄ (111) showed a weaker binding preference for *C₂H₂ than the isolated Na₂WO₄ (111) surface, with an adsorption energy of -0.54 eV versus -0.95 eV (Supplementary Fig. 41, Tables 6 and 7). In contrast, the SiO₂/Mn₃O₄ (211) favored desorption of the formed *C₂H₄ compared to pure Mn₃O₄ (211), with an adsorption energy of -0.62 eV (compared to -1.58 eV on pure Na₂WO₄ (111)). The binding energies of *CH/*CH₂ on SiO₂/Mn₃O₄ (211) and SiO₂/Na₂WO₄ (111) are -6.75 eV and -5.77 eV and -2.61 eV and 5.43 eV, respectively. These results indicate that SiO₂/Mn₃O₄ has a stronger adsorption capacity for both *CH₂ and *CH than SiO₂/Na₂WO₄. From Table S3, it was found that, regardless of whether the normal model or the inverse model, the fundamental principle remains consistent despite variations in the adsorption energy values of intermediates. For instance, Na₂WO₄/SiO₂ demonstrated a notably weaker binding preference for *C₂H₂ compared to the isolated Na₂WO₄ (111) surface, with an adsorption energy of -0.82 eV versus -0.95 eV for Na₂WO₄/SiO₂ and Na₂WO₄ (Table 3). Conversely, SiO₂/Mn₃O₄ (211) favored desorption of the formed *C₂H₄ compared to pure Mn₃O₄ (211), with an adsorption energy of -1.10 eV (compared to -1.58 eV on pure Na₂WO₄ (111)). The binding energies of *CH/*CH₂ on Mn₃O₄/SiO₂ and Na₂WO₄/SiO₂ are -8.75 eV and -

5.91 eV and -5.06 eV and -3.24 eV, respectively. These results indicate that $\text{Mn}_3\text{O}_4/\text{SiO}_2$ has a stronger adsorption capacity for both $^*\text{CH}_2$ and $^*\text{CH}$ than $\text{Na}_2\text{WO}_4/\text{SiO}_2$.

Based on the aforementioned literature and the comparison of computational results, it is reasonable to believe that the anti-model in the present work is feasible.

Original:

Supplementary information:

In addition, the different potential adsorption sites of intermediates on $\text{SiO}_2/\text{Mn}_3\text{O}_4$ (211) and $\text{SiO}_2/\text{Na}_2\text{WO}_4$ (111) were also considered to illustrate the effect of SiO_2 (Figure S39, S40 and S41; Table S6; and Table S7).

Revised in the Supplementary information: (Page 6)

Specifically, a small SiO_2 cluster (Si_8O_4) was deposited on four-layer 2×2 Mn_3O_4 (211) and Na_2WO_4 (111) surfaces (Supplementary Fig. 36). The potential adsorption sites of intermediates on $\text{SiO}_2/\text{Mn}_3\text{O}_4$ (211) and $\text{SiO}_2/\text{Na}_2\text{WO}_4$ (111) were also analyzed to elucidate the influence of SiO_2 (Supplementary Fig. 36, 39, 40 and 41; Supplementary Tables 6 and 7).

3. Furthermore, in the way the DFT calculations are reported now, they are not reproducible. For example the version of VASP is missing and there is no specifications of the used pseudopotentials. Additionally, the specified thickness of the vacuum layer as “approximately 15 Å” is needlessly vague for these calculations. Also connected to the previous point, an archive with the final geometries of the various calculations will considerably improve the reproducibility.

Reply:

We appreciate the reviewer's insightful suggestions. The details in the relevant computational section were added to the revised Supplementary information (on pages 4 to 5 in the revised Supplementary information).

Original:

Supplementary information:

As shown in Supplementary Figs. 35 37 and 38 and Tables S4 and S5, different

potential adsorption sites for reaction intermediates produced during the plasma catalytic conversion of CH₄ to C₂H₂ and C₂H₄ on the Mn₃O₄ (211) and Na₂WO₄ (111) surfaces were considered during the calculation of binding energies (BE).

Revised in the Supplementary information: (Pages 5 to 6)

Spin-polarized density functional theory (DFT)^{2,3} calculations were performed using the Vienna ab initio simulation package (VASP, version 6.21) code⁴. The exchange-correlation interactions between electrons were described using the Perdew–Burke–Ernzerhof (PBE) functional within the generalized gradient approximation (GGA)^{5,6}. Plane wave pseudopotentials with kinetic energy cutoffs of 420 eV⁷ for Mn₃O₄ and 500 eV⁸ for Na₂WO₄ were employed within the projector augmented wave (PAW) method, as determined by convergence tests. The Mn₃O₄ (211) and Na₂WO₄ (111) surfaces were selected as computational models based on experimental results (Fig. 1e and Supplementary Fig. 3). The DFT-calculated lattice constants for bulk Mn₃O₄ were $a = b = 5.841 \text{ \AA}$ and $c = 9.462 \text{ \AA}$, while for Na₂WO₄, $a = b = c = 9.157 \text{ \AA}$. These values were all in excellent agreement with previously reported values, $a = b = 5.840 \text{ \AA}$ and $c = 9.500 \text{ \AA}$ for Mn₃O₄⁹, and $a = b = c = 9.129 \text{ \AA}$ for Na₂WO₄^{9,10}. The Mn₃O₄ (211) (Supplementary Fig. 35 (a, b)) and Na₂WO₄ (111) (Supplementary Fig. 35 (c, d)) surfaces were modeled using a four-layer 2×2 surface slab. A vacuum layer with a thickness of ~15 Å was added to the slab to eliminate unphysical interactions between periodic images perpendicular to the surface. During geometry optimization, atoms in the bottom two layers were fixed, while all other atoms, including adsorbates, were allowed to relax until the force on each ion was less than 0.01 eV Å⁻¹. The convergence criterion for structure optimization was set to 1×10⁻⁵. Brillouin-zone integration was performed using a 2×2×1 Monkhorst-Pack grid with Methfessel-Paxton smearing ($\sigma = 0.2 \text{ eV}$). To account for on-site coulomb interactions, the electronic structure of Mn was treated using the DFT+U formalism, with a parameter U-J = 4.00 eV applied to the localized 3d states of Mn¹¹. In addition, van der Waals corrections were incorporated using the DFT-PBE-D3 method to accurately describe weak interactions with the catalyst¹².

As shown in Supplementary Fig. 35(a-d), Supplementary Tables 4 and 5, different potential adsorption sites of intermediates for the plasma-catalytic NOCM to C_2H_2 and C_2H_4 on Mn_3O_4 (211) and Na_2WO_4 (111) surfaces were considered during the calculation of binding energies (BE). The BE of each adsorbate was calculated as follow¹²⁻¹⁵:.....

4. In general, the figures in the main paper of low quality. The do not fulfill the pixel density, nor the minimal font size specified by the journal. This makes figures hard to read and interpret. The figures in the supporting information, especially S7, S8, S9, S29, are even worse sometimes being completely illegible.

Reply:

Thank you very much for your comments. We have improved the quality of figures in the manuscript and supporting information, ensuring they meet the journal's requirements.

Revised in Supplementary information:

Supplementary Fig. 7. (a) Production of C₂-C₃ hydrocarbons (C₂H₂ and C₂H₄, C₂H₆ and C₃H₈) under plasma-only conditions at various flow rates. (b) Molar fraction of hydrocarbons and CH₄

conversion under plasma-only conditions at various flow rates. (conditions: 1 bar, discharge power 17 W; experiment duration 60 min)

Supplementary Fig. 8 (a) Effect of discharge power on the production of C₂-C₃ hydrocarbons (C₂H₂ and C₂H₄, C₂H₆ and C₃H₈). (b) Effect of discharge power on molar fraction of hydrocarbons and CH₄ conversion. (Feed gas: 5 vol% CH₄/Ar, total flow rate 200 mL min⁻¹, the discharge powers were set to 14, 17, 20, and 23 W respectively.)

Supplementary Fig. 9 Effect of catalysts on CH₄ conversion and carbon deposition on catalysts.

(conditions: 1 bar, SEI 5.1 kJ L⁻¹, total flow rate 200 mL min⁻¹, discharge power 17 W; experiment

duration 60 min)

Supplementary Fig. 30 *In situ* FTIR spectra of different catalysts under plasma activation. (a) $\text{Mn}_3\text{O}_4/\text{m-SiO}_2$. (b) $\text{WMO}/\text{m-SiO}_2$. (c) $\text{Na}_2\text{WO}_4/\text{m-SiO}_2$.

5. Langmuir-Rideal (LR; see below) reactions are reported here to play an important role (“ Na_2WO_4 may facilitate the dehydrogenation of CH_x radicals to $^*\text{CH}$ and $^*\text{CH}_2$ species on the surface via the Eley-Rideal (E-R) mechanism.”). However, several atomistic dynamical simulations have reported that LR reaction rates might be overestimated compared to hot atom-mediated reaction, where a reactant would preferentially first adsorb on the surface, and not equilibrate (sufficiently) with the surface before another adsorbate is encountered and reaction occurs.

Reply:

We sincerely thank the reviewer for raising this important point regarding the potential overestimation of Langmuir-Rideal (L-R) reaction rates and the relevance of hot atom-mediated reactions (HA) under non-thermal plasma (NTP) conditions. The underlying surface mechanisms in plasma catalysis remain an active area of debate (J Phys Chem C, 2024, 128: 11196). Generally, E-R (L-R) reactions are considered to be enthalpy barrierless in plasma catalysis (Chem. Eng. J., 2024, 498: 155847.), which has led to concerns about the potential overestimation of L-R reaction rates compared to HA. Specially, at low coverage and low collision energy, HA dominates the abstraction, whereas HA and E-R cross-sections become similar when collision energy increases (J. Phys. Chem. C, 2015, 119: 3171.). Nevertheless, most of the previous studies have described plasma-catalytic nonoxidative methane coupling using L-R-based kinetic models (J. Phys. Chem. C, 2022, 126:19987., ACS Sustain. Chem. Eng. 2021, 9: 13151.).

In our study, a shielded bifunctional nanoreactor is proposed, where the mesoporous m-SiO₂ sphere support with a Debye shielding effect mitigates the excessive activation of reactants by plasma. Specifically, this shielding effect restricts the penetration of plasma-generated excited species into the internal catalyst regions, thereby limiting the formation of highly energetic radicals near the Na_2WO_4 and Mn_3O_4 active sites. It suggests that a significant fraction of radicals interacting with the catalyst surface may exist in neutral states, this interpretation remains tentative due to the inherent challenges in probing plasma-catalyst interfacial dynamics.

This results in a more complex reaction mechanism inside the m-SiO₂. Further theoretical calculations and mechanistic studies are necessary to assess the contributions of L-R and hot atom-mediated reactions. In light of this discussion, we have revised the corresponding section in the manuscript to refine the mechanistic interpretation.

Original:

Furthermore, Na₂WO₄ and Mn₂O₃ contribute at different stages of the methane coupling process. When CH₄ is pre-activated by plasma, Na₂WO₄ may facilitate the dehydrogenation of CH_x radicals to *CH and *CH₂ species on the surface via the Eley-Rideal (E-R), while Mn₂O₃ enhances the subsequent coupling of *CH and *CH₂ surface species to form C₂H₂ and C₂H₄.

Revised in the manuscript:

Notably, Na₂WO₄ and Mn₂O₃ function at distinct stages of the methane coupling pathway. Plasma pre-activates CH₄, facilitating Na₂WO₄-mediated enhancement of CH_x dehydrogenation into *CH and *CH₂ surface species, while Mn₂O₃ promotes subsequent coupling of these intermediates to form C₂H₂ and C₂H₄. (Page 4, lines 84-88)

*6. In the proposed reaction mechanism the authors state that the *CH and *CH₂ migrate from the Na₂WO₄ to the Mn sites, which is not supported by the results from the article. Consider specifying the nature of the migration: via desorption or via surface diffusion and providing supporting references to back up this claim. Especially since the authors also claim that these active sites are distributed independently, i.e., they are isolated.*

Reply:

Thank you for your insightful comment. We acknowledge the importance of the radical migration between different active sites in revealing the reaction mechanism, while the direct evidence for the migration of *CH and *CH₂ species between Na₂WO₄ and Mn₃O₄ sites remains inefficient. However, our conclusion is based on the following observations:

- Compared with $\text{Na}_2\text{WO}_4/\text{m-SiO}_2$, the adsorption species on the surface of the $\text{Na}_2\text{WO}_4\text{-Mn}_3\text{O}_4/\text{SiO}_2$ are different, as detected by the FTIR experiment.
- TEM results demonstrate the close proximity of Na_2WO_4 and Mn_3O_4 , and existing literature has reported the migration of intermediate species between two independent active sites.
- Computational analyses indicate that the adsorption energies of $^*\text{CH}$ and $^*\text{CH}_2$ on Mn_3O_4 are higher than those on Na_2WO_4 , suggesting a thermodynamic favoring their stabilization on Mn_3O_4 .

1. Catalytic behavior and radical migration:

The quasi-*in situ* DRIFTS characterization (Fig. 3e, Supplementary Figs. 31 and 32) indicates that the adsorbed species vary with different catalyst compositions. Specifically, the 1% $\text{Na}_2\text{WO}_4\text{-5% Mn}_3\text{O}_4/\text{m-SiO}_2$ sample exhibits the highest $^*\text{CH}_2$ intensity, followed by 1% $\text{Na}_2\text{WO}_4/\text{m-SiO}_2$ and 5% $\text{Mn}/\text{m-SiO}_2$.

More C=C species (1585 and 1465 cm^{-1}) can be observed on the 5% $\text{Mn}/\text{m-SiO}_2$ sample compared to 1% $\text{Na}_2\text{WO}_4/\text{m-SiO}_2$, indicating that $^*\text{CH}_2$ species should be primarily adsorbed on the Na_2WO_4 sites, whereas Mn_3O_4 sites facilitate their coupling to form C=C bonds. These findings indicate that a portion of the radicals can successfully migrate from one site to another on the catalyst surface before undergoing quenching or recombination, as also demonstrated by the variations in adsorbed intermediates with different catalyst loadings (Supplementary Figs. 31 and 32).

Fig. 3 | e, Quasi-*in situ* DRIFT spectra of the catalysts after NTP-CM treatment.

Supplementary Fig. 31 Quasi-*in situ* DRIFT spectra of different catalysts after plasma reaction in the regions of (a) 2850 - 2980 cm⁻¹ and (b) 1320 - 1630 cm⁻¹. (1) 2% Mn/m-SiO₂; (2) 5% Mn/m-SiO₂; (3) 10% Mn/m-SiO₂; (4) 0.5% Na₂WO₄/m-SiO₂; (5) 1% Na₂WO₄/m-SiO₂; (6) 2% Na₂WO₄/m-SiO₂; and (7) 5% Na₂WO₄/m-SiO₂.

Supplementary Fig. 32 Quasi-*in situ* DRIFT spectra of different catalysts after plasma reaction in the regions of (a) 2850 - 2980 cm⁻¹ and (b) 1320 to 1630 cm⁻¹. (1) 0.5% Na₂WO₄-5%Mn/m-SiO₂; (2) 1% Na₂WO₄-1%Mn/m-SiO₂; (3) 1% Na₂WO₄-2%Mn/m-SiO₂; (4) 1% Na₂WO₄-5%Mn/m-SiO₂; (5) 1% Na₂WO₄-10%Mn/m-SiO₂; (6) 5% Na₂WO₄-5%Mn/m-SiO₂.

On the other hand, if radicals did not migrate, carbon deposition would be expected

to accumulate on Na_2WO_4 sites. However, as shown in Supplementary Fig. 21, increasing Na_2WO_4 loading from 0.5% to 5% increases carbon deposition from 16.4% to 24.6%, whereas increasing Mn_3O_4 content reduces carbon deposition. This suggests that Mn_3O_4 suppresses excessive methane cracking and facilitates hydrocarbon formation via radical coupling, implying that $^*\text{CH}$ and $^*\text{CH}_2$ species migrate to Mn_3O_4 sites for further reactions.

Furthermore, limited carbon deposition is observed on the external surface of 1% Na_2WO_4 -5% $\text{Mn}_3\text{O}_4/\text{m-SiO}_2$, supporting a different reaction pathway where radicals generated on Na_2WO_4 diffuse into the mesoporous structure and react at Mn_3O_4 sites inside the spheres to form C_2 hydrocarbons (C_2H_2 and C_2H_4) (Fig. 3a).

Supplementary Fig. 21 Product selectivity (C, C₂H₆, C₂H₂ and C₂H₄, and C₃H₈), CH₄ conversion and carbon balance via (a) a single metal oxide (Mn-O or W-O) supported on m-SiO₂, (b) 5% Mn with varying loadings of Na₂WO₄ on m-SiO₂ and (c) 1% Na₂WO₄ with varying loadings of Mn on m-SiO₂ (conditions: 1 bar, SEI 5.1 kJ L⁻¹, total flow rate 200 mL min⁻¹, discharge power 17 W; experiment duration 60 min)

2. Proximity of Na₂WO₄ and Mn₃O₄ sites

High-resolution HRTEM shows that (Figs. 1d-1f, Supplementary Fig. 3) shows that Na₂WO₄ is well dispersed on the shell of m-SiO₂, while Mn₃O₄ is distributed within the cavity walls, leading to close proximity between the two active sites. It should be noted that the well dispersion of Na₂WO₄ on m-SiO₂ creates potential interface between Na₂WO₄ and Mn₃O₄ (Supplementary Fig. 3), which may facilitate the diffusion of reaction intermediates. The relatively higher activity of the 1%Na₂WO₄-5%Mn₃O₄/m-SiO₂ compared to Na₂WO₄/m-SiO₂ and Mn₃O₄/m-SiO₂ highlights a potential synergistic effect at these interfaces. Similar synergistic effects have been reported for many bifunctional catalysts in thermal catalysis, where intermediates relay between distinct active sites significantly improve overall catalytic efficiency, as the following discussions in references (Angew. Chem. Int. Ed., 2024, 63: e202407791.; ACS Catal., 2024, 14: 15553.; Angew. Chem. Int. Ed., 2024, 63: e202317852.; Appl. Catal. B- Environ., 2023, 323: 122190.).

Fig. 4 | g, Wavelet transform plots of the Mn K-edge and W K-edge.

Supplementary Fig. 3. TEM, STEM and HRTEM images of (a, d and g) 5% $\text{Mn}_3\text{O}_4/\text{m-SiO}_2$, (b, e and h) 1% $\text{Na}_2\text{WO}_4/\text{m-SiO}_2$ and (c, f and i) $\text{WMO}/\text{m-SiO}_2$. Here, isolated tungstate species particles (1.5-2 nm) are immobilized in the m-SiO₂ shell.

Angew. Chem. Int. Ed., 2024, 63: e202407791. It demonstrated a bimetallic design of Ag nanoparticles (NPs) and Pd single atoms (SAs) on ZnO for cascade CH_4 conversion to C_2H_4 . Initially, photogenerated charge carriers are generated on ZnO upon light illumination. The photogenerated holes enriched in the lattice oxygen sites of ZnO, activating the C–H bond cleavage of $^*\text{CH}_4$ to $^*\text{CH}_3$, which then undergoes C–C coupling on Pd SAs to generate C_2H_6 . Finally, the generated C_2H_6 deprotonates to $^*\text{C}_2\text{H}_5$, which migrates to the Ag sites for dehydrogenation to C_2H_4 .

Figure. a) Schematic illustration for the synthesis of the AgPd/ZnO composites. b–c) TEM image and HRTEM image of $\text{Ag}_3\text{Pd}_{0.3}/\text{ZnO}$. d) HAADF-STEM image and the elemental mapping results of $\text{Ag}_3\text{Pd}_{0.3}/\text{ZnO}$. e–f) Normalized Pd K-edge XANES spectra and FT-EXAFS spectra of $\text{Ag}_3\text{Pd}_{0.3}/\text{ZnO}$ in reference to Pd foil and PdO. g) Wavelet transform for Pd K-edge EXAFS spectra of $\text{Ag}_3\text{Pd}_{0.3}/\text{ZnO}$.

ACS Catal., 2024, 14: 15553. A cobalt–copper tandem catalyst for efficient and selective catalytic CO_2 -to-ethanol conversion. A CO spillover mechanism is identified as the cause of the high performance. In situ infrared spectroscopy reveals the formation of intermediate CO at Co sites, which undergo subsequent spillover and C–C coupling on the Cu clusters.

Figure. Characterization of the CoCu composite using high-resolution electron microscopy and X-ray absorption spectroscopy. (a, b) Representative HAADF-STEM images of the heterobinuclear CoCu composite at different magnifications, showing individual Cu/Co dual-sites composed of few metal atoms in close proximity. (c) EXAFS analysis of the Co–K edge of the CoCu and Co composites, and the references CoPc (containing Co(II)) and Co foil (containing Co(0)). (d) EXAFS analyses of the CoCu and Cu composites together with the references CuPc (containing Cu(II)) and Cu foil (containing Cu(0)), shown in Fourier-transformed space. (e) Normalized XANES spectra (Co–K edge) for the CoCu and Co composites and the references CoPc and Co foil. (f) Normalized XANES spectra (Cu–K edge) for the CoCu and Cu composites and the references CuPc and Cu foil.

3. Computational evidence supporting radical migration

The catalytic conversion involves the adsorption and migration of CH and CH₂, significantly influencing the product selectivity. DFT calculations reveal that the adsorption energy (Supplementary Tables 6 and 7) of *CH and *CH₂ on Mn₃O₄ (-5.21 eV and -3.96 eV, respectively) are significantly stronger than those on Na₂WO₄ (-3.27 eV and -3.08 eV, respectively). This suggests that *CH and *CH₂ species are thermodynamically more stable on Mn₃O₄, favoring their migration from Na₂WO₄ to Mn₃O₄ active sites.

Moreover, as the samples with Na₂WO₄ and Mn₃O₄ supported on m-SiO₂, the

adsorption energy (Supplementary Table 6 and 7) of *CH and *CH_2 for Na_2WO_4/SiO_2 remains consistent at -3.33 eV and -5.43 eV, respectively, while the values for Mn_3O_4/SiO_2 are enhanced to -6.75 eV and -5.77 eV. The presence of m- SiO_2 support enhanced the adsorption energy gap between Na_2WO_4 and Mn_3O_4 , reinforcing the possibility of radical migration.

Mechanism of radical migration

Based on these findings, we propose a reaction mechanism where radicals generated on Na_2WO_4 sites diffuse or desorb from the external surface of m- SiO_2 before migrating to Mn_3O_4 sites inside the spheres for further reactions (Fig. R2). This migration is likely facilitated by surface diffusion or gas-phase transport, enabling *CH and *CH_2 radicals to participate in C-C coupling at Mn_3O_4 active sites, thus enhancing the formation of C_2 hydrocarbons while reducing carbon deposition.

Figure R2. The schematic diagram illustrates the mechanism of radical diffusion, dehydrogenation, migration, and coupling processes.

We appreciate the reviewer's suggestion and have clarified the nature of radical migration (surface diffusion vs. desorption) in the revised manuscript while citing relevant literature to support this mechanism.

Modified in the manuscript: (Pages 22, 23 and 24, lines 399-445)

Reaction mechanisms. The enhanced catalytic performance of $Na_2WO_4-Mn_3O_4/m-SiO_2$, compared to $Na_2WO_4/m-SiO_2$ and $Mn_3O_4/m-SiO_2$, highlights the synergistic effects between Na_2WO_4 and Mn_3O_4 sites (Supplementary Fig. 21). Quasi-in situ DRIFTS characterization (Fig. 3e, Supplementary Figs. 31 and 32) indicates that W

sites promote *CH_2 generation, while Mn sites facilitate the coupling of *CH_2 to form C=C bonds.

If radicals were fully converted at Na_2WO_4 before reaching Mn_3O_4 sites within m-SiO₂, significant carbon deposition would be expected on Na_2WO_4 . However, compared to $Na_2WO_4/m-SiO_2$, the 1% Na_2WO_4 -5% $Mn_3O_4/m-SiO_2$ catalyst exhibits lower carbon accumulation (Supplementary Fig. 21) and higher *CH_2 and C=C intensities (Fig. 3e). These findings suggest that radicals initially generated on Na_2WO_4 sites undergo further transformation into *CH and *CH_2 , which subsequently migrate to Mn_3O_4 sites for coupling reactions to produce C_2H_2 and C_2H_4 .

Wavelet transforms of the EXAFS spectra at the Mn K-edge and W K-edge for WMO/m-SiO₂, 5% $Mn_3O_4/m-SiO_2$, and 1% $Na_2WO_4/m-SiO_2$ reveal similar Mn–O and W–O scattering peaks at (4.1 Å, 1.4 Å) and (3.9 Å, 1.2 Å), respectively (Fig. 4g). These findings indicate that Na_2WO_4 and Mn_3O_4 are independently distributed on the m-SiO₂ support, consistent with TEM observations (Supplementary Fig. 3). Notably, TEM analysis also demonstrates the close spatial proximity of Na_2WO_4 and Mn_3O_4 , providing potential pathways for intermediate species migration.

DFT calculations further reveal that the adsorption energies of *CH and *CH_2 on Mn_3O_4 are -5.21 eV and -3.96 eV, respectively, significantly stronger than those on Na_2WO_4 (-3.27 eV and -3.08 eV) (Supplementary Tables 6 and 7). Notably, when supported on m-SiO₂, the adsorption energy gap increases, suggesting that radicals preferentially stabilize on Mn_3O_4 rather than remaining on Na_2WO_4 (Supplementary Fig. 41). This thermodynamic preference, combined with the close proximity of two active sites, indicates that *CH and *CH_2 species likely undergo surface diffusion or desorption-reabsorption migration, facilitating C-C coupling on Mn_3O_4 . Similar bifunctional catalysis mechanisms have been reported in thermal catalysis systems, where intermediate spillover between distinct active sites enhances reaction efficiency^{37,38}.

The distribution and synergy between Na_2WO_4 and Mn_3O_4 sites further facilitate the sequential activation of C-H bonds and C-C coupling within the WMO/m-SiO₂ nanoreactor. Herein, a tandem reaction mechanism is proposed for CH_4 conversion to

C₂H₂ and C₂H₄ (Fig. 3h): (I) CH₄ dissociation and CH₃ diffusion: Energetic electrons from the plasma induce CH₄ dissociation into CH_x fragments, which subsequently diffuse toward the interior of the silica sphere due to the concentration gradient across m-SiO₂. (II) Dehydrogenation at Na₂WO₄ sites (support on the channel): *CH₃ species undergo dehydrogenation at W sites, forming surface-adsorbed *CH and *CH₂ species. (III) Surface species coupling on Mn sites: *CH and *CH₂ species migrate from Na₂WO₄ to Mn₃O₄ sites, leading to C-C coupling for C₂H₂ and C₂H₄ production.

Notably, the mesoporous channels of m-SiO₂ restrict plasma penetration into the interior of m-SiO₂, thereby mitigating excessive CH₄ activation (e.g., methane cracking) and suppressing carbon deposition. This structural confinement, combined with the synergistic tandem catalysis of Na₂WO₄ and Mn₃O₄, significantly enhances the yield of C₂ products. Moreover, the shielding effect of the nanoreactor reduces plasma-induced product decomposition and recombination, further enhancing selectivity.

Next follow some minor comments:

1. The authors refer to the Eley-Rideal mechanism for surface reactions. I would prefer the name Langmuir-Rideal instead of Eley-Rideal as motivated by “Prins, R. (2018-06-01). “Eley–Rideal, the Other Mechanism”. Topics in Catalysis. 61 (9): 714–721. doi:10.1007/s11244-018-0948-8. ISSN 1572-9028”. A brief summary of the article is that Dan Eley and Eric Rideal never worked on the supposed Eley-Rideal mechanism of gas phase molecule reacting with an adsorbate, which was already proposed by Langmuir. Instead, they worked on a similar but distinct reaction mechanism between a physisorbed and a chemisorbed molecule.

Reply:

Thank you for your comment. After carefully reviewing the referenced literature, we further understand the distinction between the Langmuir-Rideal and Eley-Rideal mechanisms. In light of this clarification, we have revised the relevant statements in the manuscript to accurately reflect the plasma catalytic process.

Original:

Furthermore, Na_2WO_4 and Mn_2O_3 contribute at different stages of the methane coupling process. When CH_4 is pre-activated by plasma, Na_2WO_4 may facilitate the dehydrogenation of CH_x radicals to $^*\text{CH}$ and $^*\text{CH}_2$ species on the surface via the Eley-Rideal (E-R), while Mn_2O_3 enhances the subsequent coupling of $^*\text{CH}$ and $^*\text{CH}_2$ surface species to form C_2H_2 and C_2H_4 .

Revised in the manuscript:

Notably, Na_2WO_4 and Mn_2O_3 function at distinct stages of the methane coupling pathway. Plasma pre-activates CH_4 , facilitating Na_2WO_4 -mediated enhancement of CH_x dehydrogenation into $^*\text{CH}$ and $^*\text{CH}_2$ surface species, while Mn_2O_3 promotes subsequent coupling of these intermediates to form C_2H_2 and C_2H_4 . (Page 4, lines 84-88)

2. In the Supplementary information, figures 30 and 31 are mislabeled as 27 and 28.

Reply:

Thank you very much. We have thoroughly reviewed all figure labels and corrected the mislabeling in Figures, as well as any related references throughout the manuscript.

3. The term DBD is sometimes used to refer to the device instead of the actual discharge, use DBD reactor instead. (line 226, 292)

Reply:

Thank you for your comment. We have carefully reviewed the manuscript and clarified the terminology to avoid ambiguity. Specifically, the term “DBD” has been revised to refer to the device as “DBD reactor” where appropriate. For instance:

Original:

We further compared CH_4 conversion and $\text{C}_2\text{-C}_3$ hydrocarbon distribution in the DBD with previously published literature.

Revised in the manuscript: (Page 13, lines 233-235)

A comparison of CH_4 conversion and $\text{C}_2\text{-C}_3$ hydrocarbon distribution in the DBD

reactor with previous studies is provided in Supplementary Table 2.

4. Many acronyms are never expanded or not at first occurrence: XRD, SEM, ZSM-5, TPR, XPS, DRIFTS, (NTP-)CM.

Reply:

Thank you for your comment. We have thoroughly reviewed the manuscript and ensured that all acronyms are spelled out in full upon their first mention to enhance clarity and maintain consistency.

5. units on line 146 and 153 are different, making comparison difficult.

Reply:

Thank you for your suggestion. We have ensured that the units used are consistent in the revised manuscript.

Revised in the manuscript:

Under plasma-only and plasma-catalysis conditions, the measured temperature was ~200 °C. In plasma-only mode, C₂-C₃ hydrocarbons dominated, with a maximum production of 31.2 μmol min⁻¹ at a CH₄ conversion of 33% (Supplementary Figs. 7 and 8). (Page 8, lines 145-148)

WMO/m-SiO₂ exhibited no catalytic activity for methane conversion at 250 °C in the absence of plasma. However, under NTP conditions with WMO/m-SiO₂, the production of C₂H₄ and C₂H₂ significantly increased to 30.3 μmol g⁻¹ min⁻¹ (12.8 μmol g⁻¹ min⁻¹ for C₂H₄ and 17.5 μmol g⁻¹ min⁻¹ for C₂H₂), surpassing WMO/SiO₂ and WMO/ZSM-5 by factors of ~5 and 3.4, respectively (Fig. 1c). With the WMO/m-SiO₂ nanoreactor, the proportion of unsaturated hydrocarbons in the C₂-C₃ range increased significantly from 17.7% (plasma-only) to 42.3% (Figure 1c and Supplementary Figure 9). Simultaneously, the total yield of C₂H₄ increased markedly from 2.6 μmol min⁻¹ to 6.4 μmol min⁻¹, while C₂H₂ increased from 3.1 μmol min⁻¹ to 8.8 μmol min⁻¹. (Page 8, lines 153-162)

6. The authors state that C_2H_2 and C_2H_4 fractions are quadrupled for WMO/m-SiO₂, while figure 1c does not support this. There one sees a change from ~20% to ~40%, which would suggest a doubling instead of quadrupling.

Reply:

Thank you for your comment. Upon reviewing the data, we acknowledge that the original statement inaccurately described the increase in the fractions of C_2H_2 and C_2H_4 . We have revised the corresponding text in the manuscript as follows:

Revised in the manuscript:

Furthermore, as Na₂WO₄ and Mn₃O₄ particles are encapsulated within m-SiO₂ nanospheres, the yield of C_2H_2 and C_2H_4 decreased from 7.0% (WMO/m-SiO₂) to 4.5% (Na₂WO₄-Mn₃O₄, denoted as WMO). This trend implies that converting an equivalent amount of methane leads to more carbon deposition when fewer encapsulated Na₂WO₄ and Mn₃O₄ particles are present (Fig. 2d and Supplementary Fig. 20). (Page 12, lines 213-217)

7. Figure 3h references on line 380 does not exist.

Reply:

We corrected the labeling of the referenced figures in the manuscript, updating from “3h” to “3g”.

8. In the conclusion (line 422), the author mentions the potential of combinations with technical advances to produce competitive reactor design, but mentions no examples nor references any works. Without those, this statement is too general and vague and does not support the narrative.

Reply:

Thank you for your comment. We have revised the manuscript as follows:

Original:

This promising catalyst design strategy represents a significant advancement in plasma catalysis for methane upgrading. Furthermore, when combined with other

technological advancements, such as new reactor designs, this approach has the potential to provide a cost-competitive approach for directly converting methane to unsaturated valuable light olefins.

Revised in the manuscript:

This catalyst design strategy offers significant potential for advancing plasma catalysis. It enables highly selective and directional conversion, improving the energy efficiency of plasma-catalytic systems and paving the way for a high value route to transform methane into unsaturated light olefins under mild conditions. (Page 25, lines 466-470)

9. Is the added energy in the thermal catalysis case similar to the plasma + catalyst without heating case? In other words, what is the efficiency of the various ways of adding energy? It would be a further indication of the efficiency of the plasma-assisted catalytic process compared to the thermal catalytic process.

Reply:

Thank you for your comment. The mechanisms of energy delivery during thermal and plasma-assisted catalysis are fundamentally different, resulting in distinct energy efficiencies:

In thermal catalysis, the input energy is uniformly distributed as heat to overcome activation barriers, reactants adsorbed on catalytic sites undergo a series of reactions and obtain the product driven by temperature, with the efficiency determined by factors such as heat loss of the reactor, the activity of the active sites and the activation rate of the reactants bonds.

In plasma-assisted catalysis, the input energy can initially excite and accelerate electrons in the discharge region, following the excitation of gaseous molecules through collisions, thereby initiating chemical reactions.

The energy efficiency of the plasma-assisted catalytic process compared to the thermal catalytic process for C₂H₄ production is calculated as follows:

$$f_{thermal} = \frac{Q_{C_2H_4}}{\Delta H + C_{CH_4}} \quad (R1)$$

$$f_{plasma} = \frac{Q_{C_2H_4}}{W_{discharge} + Q_{CH_4}} \quad (R2)$$

Where the $Q_{C_2H_4}$ and Q_{CH_4} are the calorific value of the C_2H_4 and CH_4 , ΔH is the enthalpy change of the reaction ($2CH_4(g) = C_2H_4(g) + 2H_2(g)$, $\Delta H_{200^\circ C} = 206 \text{ kJ/mol.}$), $W_{discharge}$ is the power of the plasma discharge. Due to the difficulty in accurately estimating heat transfer and heat utilization efficiency in thermal catalysis calculations, we instead use ΔH , which represents a slightly lower value than the actual thermal power supply. In contrast, $W_{discharge}$ accounts for the total energy used in the plasma catalytic process, including both the energy required to initiate various reactions and the thermal effect.

In a study published in *Science* (2014, 344: 616–619), Guo et al. investigated the nonoxidative coupling of methane using an Fe/SiO₂ catalyst. They achieved a maximum methane conversion of 48.1% and an ethylene selectivity of 48.4% at 1100 °C. Under plasma-catalyzed conditions in the present study, the reaction achieved an 18% selectivity for ethylene and acetylene, a 39% methane conversion, and a C_2H_4 and C_2H_2 yield of $15 \mu\text{mol min}^{-1}$ under milder conditions. Comparatively, the thermocatalytic process for C_2H_4 production exhibits an energy efficiency of 35%, whereas the plasma-catalyzed process demonstrates an efficiency of 1.7%. Nonetheless, a key advantage of plasma catalysis is its ability to overcome thermodynamic limitations, allowing the reaction to proceed at atmospheric pressure while reducing equipment costs. Additionally, plasma catalysis can utilize electricity from renewable sources, offering the potential for lower overall operational costs.

10. Only adsorption energies are calculated, not barriers. A comment in the paper about the approximation made here (barrier height scales with adsorption energies) should be mentioned. If literature results for a (few) barrier(s) in this work exist, they should be discussed.

Reply:

We appreciate the reviewer's insightful suggestions, which are crucial for

enhancing the quality of our manuscript. In this work, the Gibbs free energy (G , the detailed calculations of Gibbs free energy are provided in the supplementary material) was calculated after getting the adsorption energy of reaction intermediates produced during the plasma catalytic conversion of CH_4 to C_2H_2 and C_2H_4 on the Mn_3O_4 (211) and Na_2WO_4 (111). Here, the change in free energy (ΔG) was calculated to evaluate the degree of difficulty of the elementary reaction. The calculated Gibbs free energy profiles of CH_4 dehydrogenation and intermediate active species coupling with the Mn_3O_4 (211) and Na_2WO_4 (111) catalysts are shown in Fig. 4c, d and e. In addition, the reaction energy barriers for $^*\text{CH}_3 \rightarrow ^*\text{CH}_2 + ^*\text{H}$ and $^*\text{CH}_2 \rightarrow ^*\text{CH} + ^*\text{H}$ were also calculated (see Supplementary Table 8 in the Supplementary information, page 62), as these two elementary reactions represent the most difficult steps in the CH_4 dehydrogenation process to produce CH_3 , CH_2 , and CH species on the Na_2WO_4 (111) and Mn_3O_4 (211) surfaces, based on the Gibbs free energy change profiles of CH_4 dehydrogenation (Fig. 4c). The additional discussion according to the results from DFT calculations has been added into the revised manuscript.

Supplementary Table 8. DFT calculated activation energy (E_a) on the Mn_3O_4 (211) and Na_2WO_4 (111) models.

Elementary reactions	Mn_3O_4 (211)	Na_2WO_4 (111)
	E_a (eV)	E_a (eV)
$^*\text{CH}_3 \rightarrow ^*\text{CH}_2 + ^*\text{H}$	1.93	0.97
$^*\text{CH}_2 \rightarrow ^*\text{CH} + ^*\text{H}$	2.02	0.69

Original:

In the manuscript:

The calculated Gibbs free energy profiles of CH_4 dehydrogenation and intermediate active species coupling with the Mn_3O_4 (211) and Na_2WO_4 (111) catalysts are shown in Fig. 4c, d and e. For the CH_4 dehydrogenation process (Fig. 4c), the Na_2WO_4 (111) surface can facilitate CH_4 dehydrogenation to produce CH_3 , CH_2 , and CH species. The most difficult step is CH_3 dehydrogenation, for which the uphill energy is 0.22 eV, followed by CH_2 dehydrogenation, for which the uphill energy is 0.19 eV on the Na_2WO_4 (111) surface. In contrast, CH_4 dehydrogenation on the Mn_3O_4 (211)

surface has a much greater uphill energy (0.73 eV and 0.75 eV) for producing CH₂ and CH species, respectively.

Supplementary information:

The change in free energy (ΔG) was calculated using the following equation^{13,14,15}:

$$\Delta G = \Delta E + \Delta ZPE - T\Delta S \quad (10)$$

Where ΔE is the binding energy of the adsorbed species.

Revised in the manuscript and Supplementary information:

Manuscript: (Page 20, lines 356-371)

The calculated Gibbs free energy change profiles for CH₄ dehydrogenation and intermediate active species coupling on the Mn₃O₄ (211) and Na₂WO₄ (111) catalysts are shown in Fig. 4c, d and e. For CH₄ dehydrogenation (Fig. 4c), the Na₂WO₄ (111) surface can facilitate CH₄ dehydrogenation to produce *CH₃, *CH₂, and *CH species. The most challenging step was *CH₃ dehydrogenation, with an uphill energy of 0.22 eV, followed by *CH₂ dehydrogenation, with an uphill energy of 0.19 eV on the Na₂WO₄ (111) surface. In contrast, CH₄ dehydrogenation on the Mn₃O₄ (211) surface required significantly higher uphill energy (0.73 eV and 0.75 eV) for producing *CH₂ and *CH species, respectively. In addition, the reaction energy barriers for *CH₃ → *CH₂ + *H and *CH₂ → *CH + *H were calculated (Supplementary Table 8). These two elementary reactions were identified as the rate-determining steps in CH₄ dehydrogenation to generate *CH₃, *CH₂, and *CH species on both surfaces (Fig. 4c), with energy barriers of 0.97 eV and 0.69 eV on Na₂WO₄ (111), significantly lower than those on Mn₃O₄ (211) (1.93 eV and 2.02 eV). This indicates that the Na₂WO₄(111) surface is more effective in facilitating CH₄ dehydrogenation, consistent with experimental results (Supplementary Figure 21).

Supplementary information: (Page 7)

The change in free energy (ΔG) was calculated using the following equation¹⁴⁻¹⁶:

$$\Delta G = \Delta E + \Delta ZPE - T*\Delta S \quad (10)$$

Where ΔE is the binding energy of the adsorbed species.

The climbing image nudged elastic band (CI-NEB) method, implemented in VASP, was used in conjunction with the built-in Dimer method to identify transition states of chemical reactions. This approach has proven highly effective for determining activation energies. The activation energy (E_a) of a chemical reaction is defined as the energy difference between the transition state (TS) and the initial state (IS), while the reaction energy (ΔE) is the energy difference between the final state (FS) and the initial state. The activation energy and the reaction energy were calculated using the following equations^{12,16}.

$$E_a = E_{TS} - E_{IS} \quad (11)$$

$$\Delta E = E_{FS} - E_{IS} \quad (12)$$

where E_{IS} , E_{TS} and E_{FS} are the total energies of the initial state, transition state and final state, respectively.

11. The DFT method section in the main paper should mention PBE-D3 (like the SI), instead of only PBE (it has a dramatic influence on the results).

Reply:

We appreciate the reviewer's insightful suggestions. The relevant computational details have been added in the revised Supplementary information (on pages 4 to 5)

Original:

As shown in Supplementary Figs. 33 to 40 and Tables S4 and S5, different potential adsorption sites for reaction intermediates produced during the plasma catalytic conversion of CH₄ to C₂H₂ and C₂H₄ on the Mn₃O₄ (211) and Na₂WO₄ (111) surfaces were considered during the calculation of binding energies (BE).

Revised in the Supplementary information: (pages 5 to 6)

Spin-polarized density functional theory (DFT)^{2,3} calculations were performed using the Vienna ab initio simulation package (VASP, version 6.21) code⁴. The exchange-correlation interactions between electrons were described using the Perdew–Burke–Ernzerhof (PBE) functional within the generalized gradient approximation

(GGA)^{5,6}. Plane wave pseudopotentials with kinetic energy cutoffs of 420 eV⁷ for Mn₃O₄ and 500 eV⁸ for Na₂WO₄ were employed within the projector augmented wave (PAW) method, as determined by convergence tests. The Mn₃O₄ (211) and Na₂WO₄ (111) surfaces were selected as computational models based on experimental results (Fig. 1e and Supplementary Fig. 3). The DFT-calculated lattice constants for bulk Mn₃O₄ were $a = b = 5.841 \text{ \AA}$ and $c = 9.462 \text{ \AA}$, while for Na₂WO₄, $a = b = c = 9.157 \text{ \AA}$. These values were all in excellent agreement with previously reported values, $a = b = 5.840 \text{ \AA}$ and $c = 9.500 \text{ \AA}$ for Mn₃O₄⁹, and $a = b = c = 9.129 \text{ \AA}$ for Na₂WO₄^{9,10}. The Mn₃O₄ (211) (Supplementary Fig. 35 (a, b)) and Na₂WO₄ (111) (Supplementary Fig. 35 (c, d)) surfaces were modeled using a four-layer 2×2 surface slab. A vacuum layer with a thickness of ~15 Å was added to the slab to eliminate unphysical interactions between periodic images perpendicular to the surface. During geometry optimization, atoms in the bottom two layers were fixed, while all other atoms, including adsorbates, were allowed to relax until the force on each ion was less than 0.01 eV Å⁻¹. The convergence criterion for structure optimization was set to 1×10⁻⁵. Brillouin-zone integration was performed using a 2×2×1 Monkhorst-Pack grid with Methfessel-Paxton smearing ($\sigma = 0.2 \text{ eV}$). To account for on-site coulomb interactions, the electronic structure of Mn was treated using the DFT+U formalism, with a parameter U-J = 4.00 eV applied to the localized 3d states of Mn¹¹. In addition, van der Waals corrections were incorporated using the DFT-PBE-D3 method to accurately describe weak interactions with the catalyst¹².

As shown in Supplementary Fig. 35(a-d), Supplementary Tables 4 and 5, different potential adsorption sites of intermediates for the plasma-catalytic NOCM to C₂H₂ and C₂H₄ on Mn₃O₄ (211) and Na₂WO₄ (111) surfaces were considered during the calculation of binding energies (BE). The BE of each adsorbate was calculated as follow¹²⁻¹⁵:.....

Reviewer #2 (Remarks to the Author):

I co-reviewed this manuscript with one of the reviewers who provided the listed reports.

This is part of the Nature Communications initiative to facilitate training in peer review and to provide appropriate recognition for Early Career Researchers who co-review manuscripts.

Reviewer #3 (Remarks to the Author):

This is a thorough work on the non-oxidative coupling of methane via plasma-catalysis. The main novelty of the work lies on the design of catalytic materials, wherein mesoporous SiO₂ particles with hollow structures are loaded with Na₂WO₄ within the SiO₂ channels and Mn₃O₄ in the cavities of SiO₂. A range of experimental, characterisation and computational methods are used to obtain insights on the performance of these materials. It is discussed that Na₂WO₄ enhances the conversion of CH₄ and CH₃, Mn₃O₄ promotes the coupling of surface CH and CH₂ to C₂H₂ and C₂H₄, and SiO₂ prohibits the plasma from reaching in the cavity. The work is very interesting and novel in that it presents a catalyst design approach targeted at a plasma-catalysis system, in contrast to most works in the field that follow design approaches typical for conventional heterogeneous catalysis. Nonetheless, below a range of comments are presented that should be addressed by the authors in a revised version before consideration for publication can be made.

1. In all figures of the manuscript the selectivity of C₂H₂ and C₂H₄ is presented as a lump. Considering that the main design target of the presented materials is the enhancement of the yield/selectivity of C₂ species other than ethane, it is not clear why the manuscript omits details on this regard. Given the GC and MS experimental details provided, the authors clearly can separate the two species C₂H₂ and C₂H₄ in the reaction products and provide detailed selectivities. Supplementary Fig. 21b is the only figure in the entire manuscript that does show separate traces for C₂H₂ and C₂H₄. It would be necessary to update all figures in the main text and Supplementary information to show the selectivities for C₂H₂ and C₂H₄ separately. There is a substantial difference if the presented lumped yields have been achieved through a high

selectivity in C₂H₂ versus C₂H₄. There are numerous plasma-based methods already presented in literature that have achieved very high selectivity to C₂H₂, with many of these methods not even requiring a catalyst, for example those based on nanosecond pulsed discharges. On the contrary, achieving a high selectivity to C₂H₄ has to date been a major challenge. The discussion would need to be updated to reflect the above, depending on whether it is C₂H₂ or C₂H₄ that is primarily formed.

Reply:

Thank you for your valuable comment. We have updated the Figures in the main text and Supplementary information to separately display the selectivity of C₂H₂ and C₂H₄. While our study initially presented the selectivity as a lump due to the synchronized proportions of C₂H₂ and C₂H₄ formation, we recognize the importance of distinguishing between these two products to enhance the clarity of the results and their interpretation. Therefore, we have redrawn all the related figures (Figs. 1c, 2d, 3a and 3b and Supplementary Figs. 7, 8, 9, 14, 20, 29 and 33), showing the selectivity of both C₂H₂ and C₂H₄ in the revised manuscript.

Revised in the manuscript:

Fig. 1c shows the distributions of C₂H₄ and C₂H₂ within the C₂-C₃ range. A synchronized increase in C₂H₂ and C₂H₄ proportions was observed, which can be attributed to the closely aligned energetic thresholds of electron-induced CH₄ conversion into CH₂ and CH species¹³. WMO/m-SiO₂ exhibited no catalytic activity for methane conversion at 250 °C in the absence of plasma. However, under NTP conditions with WMO/m-SiO₂, the production of C₂H₄ and C₂H₂ significantly increased to 30.3 μmol g⁻¹ min⁻¹ (12.8 μmol g⁻¹ min⁻¹ for C₂H₄ and 17.5 μmol g⁻¹ min⁻¹ for C₂H₂), surpassing WMO/SiO₂ and WMO/ZSM-5 by factors of ~5 and 3.4, respectively (Fig. 1c). With the WMO/m-SiO₂ nanoreactor, the proportion of unsaturated hydrocarbons in the C₂-C₃ range increased significantly from 17.7% (plasma-only) to 42.3% (Figure 1c and Supplementary Figure 9). Simultaneously, the total yield of C₂H₄ increased markedly from 2.6 μmol min⁻¹ to 6.4 μmol min⁻¹, while C₂H₂ increased from 3.1 μmol min⁻¹ to 8.8 μmol min⁻¹. (Pages 8, lines 150-162)

Fig. 1 | c, Production rates and molar fractions of C₂H₄ and C₂H₂ within C₂-C₃ hydrocarbons (conditions: 1 bar, specific energy input (SEI) 5.1 kJ L⁻¹, where SEI is defined as plasma discharge power divided by gas flow rate; Feed gas: 5 vol% CH₄/Ar, total flow rate 200 mL min⁻¹, discharge power 17 W; Experiment duration 60 min).

Fig. 2 | d, Yields of C₂H₂ and C₂H₄ (defined as the product of CH₄ conversion and the selectivity of C₂H₂ and C₂H₄) and equivalent carbon deposition rate (ECR, defined as the solid carbon selectivity on the catalyst divided by the methane conversion) (conditions: 1 bar, SEI 5.1 kJ L⁻¹, total flow rate 200 mL min⁻¹, discharge power 17 W; experiment duration 60 min).

Fig. 3 | Performance of Mn and W species. **a**, Selectivity of products (carbon deposited on the catalyst, C₂H₄ and C₂H₂) and CH₄ conversion for plasma-only and plasma-catalysis systems (conditions: 1 bar, SEI 5.1 kJ L⁻¹, total flow rate 200 mL min⁻¹, discharge power 17 W; experiment duration 60 min). **b**, Selectivity of carbon deposited on the catalyst and selectivity of C₂H₄ and C₂H₂ for reduced WMO/m-SiO₂ (reduced at 450 °C with H₂).

Supplementary Fig. 7 (a) Production of C₂-C₃ hydrocarbons (C₂H₂ and C₂H₄, C₂H₆ and C₃H₈) under plasma-only conditions at various flow rates. (b) Molar fraction of hydrocarbons and CH₄ conversion under plasma-only conditions at various flow rates. (conditions: 1 bar, discharge power 17 W; experiment duration 60 min)

Supplementary Fig. 8 (a) Effect of discharge power on the production of C₂-C₃ hydrocarbons (C₂H₂ and C₂H₄, C₂H₆ and C₃H₈). (b) Effect of discharge power on molar fraction of hydrocarbons and CH₄ conversion. (Feed gas: 5 vol% CH₄/Ar, total flow rate 200 mL min⁻¹, the discharge powers were set to 14, 17, 20, and 23 W respectively.)

2. In relation to the above comment, in lines 207 onwards the authors state that “... the yield of ethylene (C₂H₄) decreases from 7.7% (WMO/m-SiO₂) to 4.7% (Na₂WO₄-Mn₃O₄, denoted as WMO)” referring to Fig. 2b and Supplementary Fig. 19. Firstly, the authors likely mean Fig. 2d and not Fig. 2b. In all cases, both in Fig. 2d and Supplementary Fig. 19, where these results are presented, only lumps of ethylene and acetylene are shown.

Reply:

Thank you for your comment. We appreciate you bringing this to our attention. We have corrected the reference in the corresponding section.

Revised text in the manuscript:

Furthermore, as Na₂WO₄ and Mn₃O₄ particles are encapsulated within m-SiO₂ nanospheres, the yield of C₂H₂ and C₂H₄ decreased from 7.0% (WMO/m-SiO₂) to 4.5% (Na₂WO₄-Mn₃O₄, denoted as WMO). This trend implies that converting an equivalent amount of methane leads to more carbon deposition when fewer encapsulated Na₂WO₄ and Mn₃O₄ particles are present (Fig. 2d and Supplementary Fig. 20). (Page 12, lines 213-217)

Fig. 2 | d, Yields of C₂H₂ and C₂H₄ (defined as the product of CH₄ conversion and the selectivity of C₂H₂ and C₂H₄) and equivalent carbon deposition rate (ECR, defined as the solid carbon selectivity on the catalyst divided by the methane conversion) (conditions: 1 bar, SEI 5.1 kJ L⁻¹, total flow rate 200 mL min⁻¹, discharge power 17 W; experiment duration 60 min).

3. The manuscript does not determine anywhere what was the power applied, as determined by the Lissajous plots shown in Supplementary figures 11-12. There is only a mention of a specific energy input (SEI) value in the Methods section. What was the power and flowrates that resulted in this SEI? Were these parameters constant for all experiments carried out? There are a range of different cases studied in this work, so for clarity it would be more appropriate if all figure captions explicitly stated the applied power, flowrate and SEI.

Reply:

Thank you for your comment. We have added the discharge parameters to the “Experimental Section” and explicitly included them in the figure captions for clarity. Plasma-only experiments were performed under various discharge parameters. For all other experiments, consistent conditions were maintained as follows:

Fig. 1 | Characterization and catalytic performance. a, TEM image and particle size distribution of m-SiO₂. b, SEM image of WMO/m-SiO₂. c, Production rates and molar fractions of C₂H₄ and C₂H₂ within C₂-C₃ hydrocarbons (conditions: 1 bar, specific energy input (SEI) 5.1 kJ L⁻¹, where SEI is defined as plasma discharge power divided by gas flow rate; Feed gas: 5 vol% CH₄/Ar, total flow rate 200 mL min⁻¹, discharge power 17 W; Experiment duration 60 min). d-e, TEM and

HRTEM images of WMO/m-SiO₂. **f**, EDS line scans (from Point 1 to Point 2) from Fig. 1(d). **g**, Pore size distributions of m-SiO₂, WMO/m-SiO₂, WMO/ZSM-5 and WMO/SiO₂. **h**, C₂-C₃ production and methane conversion for catalysis-only, plasma-only and plasma-catalysis systems.

Fig. 2 | Effect of catalyst location m-SiO₂ on NTP-CM performance. **a**, TEM image and EDS spectra of spent catalysts In-m-SiO₂, Out-m-SiO₂, and Both-m-SiO₂ (after 60 min of reaction). **b**, Schematic illustration of the role of m-SiO₂ and the effect of catalyst position on methane coupling. **c**, Mn³⁺/(Mn³⁺+Mn²⁺) ratio for fresh and spent catalysts. **d**, Yields of C₂H₂ and C₂H₄ (calculated as CH₄ conversion multiplied by C₂H₂ and C₂H₄ selectivity) and equivalent carbon deposition rate (ECR, defined as carbon deposition selectivity divided by methane conversion) (conditions: 1 bar, SEI 5.1 kJ L⁻¹, total flow rate 200 mL min⁻¹, discharge power 17 W; experiment duration 60 min).

Supplementary Fig. 7 (a) Production of C₂-C₃ hydrocarbons (C₂H₂ and C₂H₄, C₂H₆ and C₃H₈) under plasma-only conditions at various flow rates. (b) Molar fraction of hydrocarbons and CH₄ conversion under plasma-only conditions at various flow rates. (conditions: 1 bar, discharge power 17 W; experiment duration 60 min)

4. In continuation to the above comment, and more importantly, there is no discussion at all in the manuscript on what is the energy efficiency or energy cost of the process. The obtained results need to be compared on the one hand with those from literature not only in terms of yield achieved, but also in terms of the energy cost for the conversion of CH₄ and the production of C₂H₄. Additionally, the energy efficiency of the process needs to be presented and discussed, comparing specifically the energy cost of the plasma generation to the reaction enthalpy of the overall methane conversion.

Reply:

Thank you for your comment. The comparison specifically the energy cost of the plasma generation to the reaction enthalpy of the overall methane conversion is added to the Supplementary Information Supplementary Table 2. In previous literature C₂H₄ and C₂H₂ is presented as a lump, so C₂H₄ and C₂H₂ will be presented as a lump in the comparison. The energy cost (EC_{CH₄}) for CH₄ conversion to C₂H₂, C₂H₄, C₂H₆ and

C₃H₈ were calculated as follows:

$$EC_{CH_4} = \frac{W_{discharge}}{N_{CH_4}} \quad (R3)$$

where $W_{discharge}$ is the discharge power under plasma-catalytic NOCM, N_{CH_4} represents the molar of CH₄ converted to hydrocarbon product (C₂H₂, C₂H₄, C₂H₆, or C₃H₈).

Revised in the manuscript:

A comparison of CH₄ conversion and C₂-C₃ hydrocarbon distribution in the DBD reactor with previous studies is provided in Supplementary Table 2. The 1% Na₂WO₄-5% Mn₃O₄/m-SiO₂ (WMO/m-SiO₂) catalyst demonstrated the high selectivity for C₂H₂ and C₂H₄, while maintaining competitive methane conversion. Among reported studies, this work achieved a lower energy cost (EC) for CH₄ conversion (6.8 MJ/mol), demonstrating the effectiveness of the catalyst in plasma-catalytic NOCM reaction.

(Pages 13, lines 233-239)

Supplementary Table 2 Comparison of CH₄ conversion (X), the fraction of C₂H₂ and C₂H₄ within C₂-C₃ hydrocarbons (F), the selectivity of C₂H₂ and C₂H₄ (R), the energy cost for the conversion of CH₄ (EC_{CH₄}) in the DBD reactor between this study and the literature.

Feed Gas	Power (W)	Samples	X (%)	F (%)	R (%)	EC _{CH₄} (MJ/mol)	Source
5% CH ₄ /Ar	17	WMO/m-SiO ₂	34	42.3	22.5	6.8	This work
CH ₄	25	Plasma only	~25	~18.5	~15	6.7	Wang et al. ⁹
CH ₄	45	Plasma only	25.2	25	12	4.8	Xu et al. ¹⁰
25% CH ₄ /Ar	21	Plasma only	29.2	28.6	20.1	12.1	Wang et al. ¹¹
CH ₄	45	Plasma only	~25	~26	~13	4.8	Liu et al. ¹²
9% CH ₄ /Ar	12	γ-Al ₂ O ₃	~15	28.6	~13	21.7	Song et al. ¹³
9% CH ₄ /Ar	12	α-Al ₂ O ₃	~11	34.3	~17	29.6	
CH ₄	38	Ru/TiO ₂	32.2	12.4	4.9	5.3	Kim et al. ¹⁴
CH ₄	38	Ru/TiO ₂	35.4	10.5	4.6	4.8	
CH ₄	--	Pt/γ-Al ₂ O ₃	11	0.3	3.9	--	Kim et al. ¹⁵

CH ₄	--	Co/-Al ₂ O ₃	28.7	15.9	5.5	--	
CH ₄	--	Pt/-Al ₂ O ₃	25.9	4.3	2.4	--	
CH ₄	50	Fe/Al ₂ O ₃	12.1	16.7	11.4	37	Indarto et al. ¹⁶
CH ₄	35	Mn/Al ₂ O ₃	10.4	23.9	12.4	15.1	
CH ₄	35	Ru/Al ₂ O ₃	9.54	23.4	14.9	16.4	
CH ₄	35	Zeolite	7.6	22	10.6	20.6	
CH ₄	50	Cu/Zeolite	15.7	26.7	13.8	14.26	
CH ₄	50	Ni/Zeolite	21.8	26.8	15.5	20.5	
CH ₄ /CO ₂	40	Plasma only	25	21.8	12	8.6	Tu et al. ¹⁷
CH ₄ /CO ₂	50	Ni/Al ₂ O ₃	31.4	14.6	12	8.6	
CH ₄ /O ₂ /Ar	--	Ag/SiO ₂	27.5	23.9	8	--	Lee et al. ¹⁸

EC_{CH_4} = discharge power/the molar amount of converted CH₄

5. The feed stream comprised of a heavily diluted mixture at a 5% CH₄ in Ar, however the role of Ar is not discussed in the manuscript. There are multiple experimental works (e.g. those of Jo et al. 10.1063/1.4818795 and 10.1016/j.ces.2015.03.019) and computational studies (e.g. that of Maitre et al. 10.1016/j.ces.2022.117731) that extensively discuss the major contribution of Ar in the activation of methane through a range of energy transfer mechanisms, including Penning ionisations and dissociations. The yields achieved are very likely affected by the presence of Argon in the reactant mixture and would be much lower in a pure CH₄ feed. Similarly, carbon deposition would be significantly different and likely more pronounced with a pure CH₄ stream. Appropriate consideration to this needs to be given in the discussion and ideally results from equivalent experiments with pure CH₄ need to be presented.

Reply:

Thank you for your comment. Ar play a critical role in the NTP-driven coupling of methane (NTP-CM) (Physics of plasmas, 2013, 20; Chemical Engineering Science, 2015, 130: 101; Chemical Engineering Science, 2022, 259: 117731.), Ar favors the

occurrence of discharge and promotes methane conversion under NTP conditions. In this study, the feed gas composition of 5% CH₄ in Ar was consistently maintained across all experiments to ensure a uniform basis for comparing the performance of different catalysts in NTP-CM. Consequently, the specific role of Ar was not explored in detail in this manuscript.

To minimize the influence of mass transfer limitations and accurately evaluate the intrinsic catalytic activity, a diluted methane feed (5% CH₄ in Ar) with a flow rate of 200 mL/min was employed. This approach aligns with experimental setups reported in recent studies (e.g., *J. Am. Chem. Soc.*, 2023, 145, 20792; *Angew. Chem. Int. Ed.*, 2023, 62, e202307814). It is important to note that high concentrations of methane could negatively impact the selectivity and conversion due to the limited active sites of the catalyst. The addition of Ar facilitates the conversion of methane with a more intuitive observation of the selective modulation effect. Ar was chosen as the carrier gas due to its distinct mass spectrometry signals (e.g., $m/z = 20$ and $m/z = 40$), which allow for real-time monitoring of gas composition trends without interference from overlapping signals, such as those of N₂ ($m/z = 28$) and C₂H₄ ($m/z = 28$).

Regarding your suggestion for experiments with pure CH₄, we conducted equivalent experiments using flow rates of 200 mL min⁻¹ for pure CH₄. The selectivity of C₂H₂ and C₂H₄ at 5% CH₄/Ar increased from 8.5% and 9.9% to 17.9% and 24.45% relative to plasma only conditions, respectively, and the production of C₂H₂ and C₂H₄ increased from 2.6 μmol min⁻¹ and 3.1 μmol min⁻¹ to 6.4 μmol min⁻¹ and 8.8 μmol min⁻¹ relative plasma conditions, respectively. Under pure CH₄ conditions, the selectivity of C₂H₂ and C₂H₄ increased from 12.9% and 13.7% to 21.8% and 18.3% relative plasma only conditions, respectively, and the production of C₂H₂ and C₂H₄ increased from 5.3 μmol min⁻¹ and 5.7 μmol min⁻¹ to 12.7 μmol min⁻¹ and 10.6 μmol min⁻¹ relative plasma conditions, respectively. The results demonstrate that the presence of Ar enhances the selectivity toward C₂H₂ and C₂H₄. A similar increasing trend of C₂H₂ and C₂H₄ is observed when comparing the 5% CH₄/Ar with pure CH₄ feed. However, this is accompanied by an increase in carbon deposition on the catalyst, as illustrated in Fig. R3. This suggests that while Ar plays a crucial role in improving selectivity and

promoting radical coupling, the overall trends in hydrocarbon formation remain consistent across both feed compositions.

Figure. R3. (a) Product distribution and (b) production of C₂H₂, C₂H₄, C₂H₆ and C₃H₈ under 5% CH₄/Ar and pure CH₄ conditions, using the catalyst 1% Na₂WO₄-5% Mn₃O₄/m-SiO₂. (1 bar, SEI = 5.1 kJ L⁻¹, Flow rate = 200 mL min⁻¹ and discharge power = 17 W. Each experiment lasted 60 min)

Revised in the “Methods” section of the manuscript:

Original:

A preheated feed gas mixture (30 °C) containing 5% CH₄ in argon (total flow rate: 200 mL min⁻¹) was introduced into the reactor and allowed to flow for 60 min.

Revised:

A diluted methane feed (5 vol% CH₄ in Ar) at a total flow rate of 200 mL min⁻¹ was used to minimize mass transfer limitations and accurately evaluate the intrinsic catalytic activity. The feed gas was preheated to 30 °C, and the experiments were conducted for 60 min. (Pages 27 and 28, lines 515-519)

6. *Supplementary figure 12 depicting electrical signals shows that the current has a peculiar triangular shape. Normally, a sinusoidal signal with superimposed spikes due to microdischarges is expected. What is the reason for this signal profile? The manuscript Supplementary information should provide details on how the current signal was measured. For completeness and for comparison, the Supplementary information should also show the time resolved V and Q signals, besides the Lissajous plots.*

Reply:

The current signals were acquired by measuring the voltage across a 50 Ω external resistor integrated into the circuit using a voltage probe set to average mode. When the voltage probe was set to sample or high-resolution mode, the current exhibited a sinusoidal waveform with superimposed spikes, characteristic of micro-discharges. However, these modes introduced greater fluctuations in the calculated discharge power. To minimize these fluctuations and achieve more stable and precise measurements, the average mode was selected for recording the current profile. The peculiar triangular shape of the current signal observed in this mode results from the integration of multiple current waveforms, which can be found in the literature (Plasma Science and Technology, 2018, 20(4): 044009. The European Physical Journal D, 2022, 76(5): 77. Scientific Reports, 2023, 13(1): 7394.). This average mode enabled more consistent and accurate discharge power measurements.

We have revised the description of the electrical signal measurement in the “Plasma reactor and parameter measurement” section of the Supplementary information. Additionally, we have updated the Supplementary information to include detailed explanations of the measurement methodology, time-resolved voltage (V) and charge (Q) signals for comparison, and Lissajous plots, providing a more comprehensive analysis of the electrical characteristics.

Revised in Supplementary information:

1.3 Plasma reactor and electrical measurements

The reaction was conducted in a dielectric barrier discharge (DBD) plasma reactor with a discharge length of 50 mm and a discharge gap of 2 mm (see Supplementary Fig. 1). A stainless steel mesh wrapped around a quartz tube (with an inner diameter of 8 mm and a wall thickness of 2 mm) served as the high-voltage electrode. A corundum tube (inner diameter: 2 mm, wall thickness: 1 mm), positioned along the axis of the quartz tube, housed a 2-mm diameter stainless-steel rod that served as the ground electrode.

A high-voltage AC power source (CTP-2000K, Nanjing Suman Electronic Co. Ltd) was used to power the DBD reactor. The applied voltage of the DBD was measured by a high-voltage probe (Tektronix, P6015A), while the current was recorded by measuring the voltage drop across an integrated resistor (50Ω) using a voltage probe. An external capacitor ($0.47 \mu\text{F}$) was used to measure the charge formed during the discharge. All electrical signals were recorded by a digital oscilloscope (SIGLENT, SDS1102X-C). The discharge power was calculated using the typical Q-U Lissajous figure method¹.

Supplementary Fig. 12 Electrical signals for plasma-only and plasma with different catalysts (WMO/m-SiO₂, WMO/ZSM-5 and WMO/SiO₂). (a) Current. (b) Applied voltage. (c) Charge (conditions: 1 bar, SEI 5.1 kJ L^{-1} , total flow rate 200 mL min^{-1} , discharge power 17 W ; experiment duration 60 min)

7. The authors stress the importance of balancing the oxygen content in the catalysts and also carry out comparative experiments with pre-reduced samples to demonstrate this. Considering that H_2 is a primary product of the coupling reaction, are the oxides expected to be eventually reduced? The plasma operation at extended periods will unavoidably lead to parasitic heating of the catalyst bed.

Reply:

Thank you for your comment. The **X-ray photoelectron spectroscopy (XPS) and X-ray Diffraction (XRD) Analysis of Spent Catalyst show meaningful information on this issue:** As shown in Supplementary Fig. 24 and 25, a comparison of the XPS spectra of the fresh and spent catalyst reveals no significant differences in W and Mn oxidation states, indicating that the metal oxides remain stable during the reaction. Based on these observations, we conclude that the metal oxides will not fully be reduced under the reaction conditions for the following reasons:

Supplementary Fig. 24 XPS spectra in the W 4f and Mn 2p regions of the fresh and spent catalysts. (a, b) WMO/m-SiO₂. (c) 1% Na₂WO₄/m-SiO₂. (d) 5% Mn/m-SiO₂.

Supplementary Fig. 25. XPS spectra WMO/m-SiO₂ after reaction for 9 h, 14 h and 24 h. (a) W 4f region. (b) Mn 2p region.

1. Debye Shielding Effect: The plasma discharge cannot penetrate the mesopores of m-SiO₂ due to the “Debye shielding” effect, which limits radical access to the interior regions of m-SiO₂ where the metal oxides are predominantly located. This structural design minimizes the exposure of the metal oxides to reactive radicals that could potentially induce reduction. The relevant discussion in the manuscript:

EDS analysis of Particle 1 (internal) revealed weaker carbon and stronger oxygen signals compared to Particle 2 (external) (Fig. 2a and Supplementary Fig. 17). This suggests that m-SiO₂ protects Mn₃O₄ particles from direct exposure to CH₄ plasma. This shielding effect arises from the “Debye shielding” mechanism, where plasma discharge cannot penetrate the mesopores of m-SiO₂ due to their pore diameters being smaller than the Debye length (typically hundreds of nanometers)^{19,25,26}. Thus, the shielded internal cavity prevents the reduction of Mn₃O₄ particles and mitigates direct carbon deposition. This hypothesis is further supported by the observed decrease in the Mn³⁺/(Mn³⁺+Mn²⁺) ratio as more manganese oxide particles are located outside the m-SiO₂ layer (Fig. 2c and Supplementary Fig. 18). (Pages 11 and 12, lines 200-209)

Supplementary Fig. 18 XPS spectra in the Mn 2p and W 4f regions (from top to bottom) of fresh Both-m-SiO₂ and Out-m-SiO₂, as well as spent Both-m-SiO₂ and Out-m-SiO₂.

2. Catalyst Bed Temperature: Although extended plasma operation may result in parasitic heating of the catalyst bed, the measured temperature under plasma-only and plasma-catalysis was approximately 200 °C during the catalytic process, as shown in Supplementary Fig. 27. This conclusion is corroborated by H₂-TPR analysis, which demonstrates that a higher temperature is required for the reduction of the metal oxides.

Supplementary Fig. 27 H₂-TPR profiles of WMO/m-SiO₂, 1% Na₂WO₄/m-SiO₂ and 5% Mn₃O₄/m-SiO₂.

8. More details need to be provided on the isotopic labelling experiments, namely the conversion and selectivities achieved for each of the presented experiments.

Reply:

Thank you for your comment, which are crucial for enhancing the quality of our manuscript. Accordingly, we have revised the “Methods” section to provide a more comprehensive description, including the conversion of reactants and the selectivity of products under feed gas 2.5% CH₄-2.5% CD₄/Ar.

Under plasma discharge, CH₄ and CD₄ are dissociated into radicals such as CH₃, CH₂, CH, CD₃, CD₂, CD, H, and D. These radicals recombine to form isotopically labeled methane species, including CD₃H (m/z = 19), and CD₂H₂ (m/z = 18). The formation of CD₃H and CD₂H₂ occurs through the recombination of radicals, corresponding to the reactions of CD₃ with H and CD₂ with 2H, respectively. Mass spectrometric signals at m/z = 15, 18, and 19 were recorded, where m/z = 15 and m/z = 19 correspond to the concentrations of CH₄ and CD₃H, respectively. The m/z = 18 includes the contribution of CD₂H₂ and the fragment of CD₃ from CD₄. Additionally, the mixed concentrations of CH₄, CD₄ and reaction products were quantified using gas chromatography and mass spectrometry. These data were used to calculate the conversion of CH₄ and CD₄, as well as the selectivity of the reaction products (Supplementary Fig. 33).

Supplementary Fig. 33 Effect of catalysts on the molar fractions of C₂-C₃ hydrocarbons and

the conversion of CH₄ and CD₄ (conditions: 1 bar, feed gas 2.5% CH₄-2.5% CD₄/Ar, SEI 5.1 kJ L⁻¹, total flow rate 200 mL min⁻¹, discharge power 17 W; experiment duration 60 min).

Compared to that in the plasma-only system, the plasma catalytic system with the presence of WMO/m-SiO₂ catalyst demonstrated a higher production of CH₂D₂ and CHD₃ (Fig. 3f). The corresponding conversion of CD₄ and CH₄ and products shown in Supplementary Fig. 33. This suggests that WMO/m-SiO₂ promotes the activation of CH₄ and the generation of CH₃/CD₃ and CH₂/CD₂ radicals. This enriched pool of CH₂ species ultimately leads to increased dimerization into C₂H₂ and C₂H₄.

Fig. 3 | f, CH₂D₂ and C₂HD₃ species generated under plasma only and NTP-WMO/m-SiO₂ conditions (feed gas is 2.5% CH₄-2.5% CD₄/Ar, SEI = 5.1 kJ L⁻¹, Flow rate is 200 mL min⁻¹. Each experiment lasted 60 min)).

Revised in manuscript:

Isotopic labeling experiments. To investigate the relative contributions of CH₂ and CH₃ radicals to CH₄ decomposition, isotopic labeling experiments were conducted. The catalyst was first pretreated in a 5 vol% CH₄/Ar mixture (discharge power: 17 W, flow rate: 200 mL min⁻¹) for 60 min and then cooled to room temperature. Next, argon (200 mL min⁻¹) was flowed through the reactor for 10 min to purge residual gases. A gas mixture of 2.5 vol% CD₄ and 2.5 vol% CH₄ in argon (200 mL min⁻¹) was subsequently introduced into the DBD reactor for 10 min. The plasma was then switched on and sustained for 60 min, during which mass spectrometric signals at m/z = 15, 18, and 19

were monitored. The concentrations of CH₄ and CD₄ in the mixture were determined using mass spectrometry. From these measurements, the conversion of CH₄ and CD₄, as well as the selectivity of the product, was determined. Under plasma discharge conditions, CH₄ and CD₄ dissociate into radicals including CH₃, CH₂, CH, CD₃, CD₂, CD, H, and D. These radicals recombine to form isotopically labeled methane species, such as CD₃H (m/z = 19), and CD₂H₂ (m/z = 18). The formation of CD₃H and CD₂H₂ results from the recombination reactions of CD₃ with H and CD₂ with two H atoms, respectively. The signals at m/z = 15 and m/z = 19 reflect the concentrations of CH₄ and CD₃H, respectively. When analyzing the peak intensity of CD₂H₂, it is critical to consider the contribution of CD₄ fragmentation (m/z = 20), which generates CD₃ (m/z = 18). Specifically, the signal at m/z = 18 includes contributions from both CD₂H₂ and the CD₃ fragment derived from CD₄. (Page 29, lines 539-558)

9. The lifetime of the radicals while diffusing in the mesoporous SiO₂ needs to be estimated and considered in the discussion. Depending on the diffusion length, the radicals can possibly preferentially recombine before reaching the inside of the particles for catalytic reactions. Also related to this comment, the contribution of plasma phase is largely neglected in the discussion and should be considered.

Reply:

Thank you for your comments about the radical lifetime and the role of plasma-phase reactions. Regarding the lifetime of radicals and their diffusion within mesoporous SiO₂, previous numerical simulations (Catal. Today 2004, 98: 607) have shown that in the non-oxidative conversion of methane, various radicals and excited species are generated, including fragments of CH_x (x = 0–3), atomic H, and vibrationally excited CH₄. As illustrated in Fig. R4, the molar densities of C, CH, and CH₂ decrease rapidly after the termination of plasma filaments, whereas CH₃ exhibits significantly longer stability (>1 ms) compared to CH₂ (<30 ns) and CH (<5 ns). This suggests that CH₃ radicals are the most abundant and long-lived species under plasma conditions, allowing for their effective participation in subsequent catalytic reactions. Moreover, in the absence of catalysis, the formation of unsaturated C₂ hydrocarbons (e.g., C₂H₂ and

C₂H₄) is primarily limited by the recombination of CH and CH₂ radicals.

[Figure Redacted]

Figure R4. Evolution of a streamer at 300 K–100 kPa and 200 Td: (a) electron and radicals. (Catal. Today 2004, 98: 607)

TEM characterization confirms that Mn₃O₄ sites in Mn₃O₄/m-SiO₂ catalysts are well dispersed within mesoporous SiO₂ nanospheres, with a shell thickness of approximately 20 nm. Compared to m-SiO₂, the introduction of Mn₃O₄ (2% Mn₃O₄/m-SiO₂) promotes the formation of unsaturated hydrocarbons (C₂H₄, C₂H₂) (Fig. R5) and reduce the carbon deposition. Despite the fact that the internal Mn₃O₄ sites are not directly exposed to plasma, they effectively influence the plasma-driven reaction pathways, suggesting that a fraction of radicals can diffuse into the interior and undergo surface catalytic reactions at Mn₃O₄ sites. Furthermore, varying the Mn₃O₄ loading within the m-SiO₂ spheres leads to observable changes in product distribution (Supplementary Fig. 21), particularly in the C₂H₆ and C₃H₈ yields and carbon deposition, further indicating that Mn₃O₄ sites actively participate in radical-induced surface reactions, even though they are not directly exposed to plasma.

Quasi-in situ DRIFT spectra (Fig R6.) reveal that the relative intensity of adsorbed *CH₃ and *CH₂ species varies with the internal Mn₃O₄ loading, suggesting a direct correlation between radical adsorption and Mn₃O₄ site availability. Based on reported computational studies and the observed catalytic performance, CH₃ radicals, which exhibit a longer lifetime compared to other radicals, are more likely to diffuse into the mesoporous structure and undergo surface reactions at the internal Mn₃O₄ sites. This further supports the hypothesis that radical migration plays a key role in the reaction mechanism.

Figure R5. Product selectivity using different catalysts.

Figure R6. Quasi-*in situ* DRIFT spectra in regions from 2850 to 2980 cm^{-1} (a) and 1320 to 1630 cm^{-1} (b) for catalysts after NTPC-CM via 2% Mn/m-SiO₂, 5% Mn/m-SiO₂ and 10% Mn/m-SiO₂.

Regarding the contribution of the plasma phase, its primary role is to activate the C-H bond and elevate the energy level of CH₄, thus facilitating its interaction with catalytic sites. Plasma also enhances the formation of highly reactive CH_x intermediates, promoting rapid surface reactions at lower temperatures. The synergistic effect between plasma activation and catalyst-driven surface reactions is a key factor in the observed catalytic performance. While the plasma-phase recombination of radicals could limit their availability, the experimental results suggest that a sufficient fraction of active

radicals remains available for catalytic transformations, leading to the enhanced production of unsaturated hydrocarbons.

We have incorporated this discussion in the revised manuscript to address the reviewer's concerns.

Revised in Manuscript:

Although the internal Mn_3O_4 sites are not directly exposed to plasma (Supplementary Fig. 3), varying the Mn_3O_4 loading within the m- SiO_2 spheres led to observable changes in product distribution and the relative intensity of adsorbed $^*\text{CH}_3$ and $^*\text{CH}_2$ species (Supplementary Figs. 21 and 31). This suggests that CH_x radicals can diffuse or migrate at least 20 nm to reach Mn_3O_4 sites within their lifetime, enabling them to access the interior of the catalyst for subsequent reactions. (Page 16, lines 293-298)

10. There are errors in the labelling of supplementary figures 30-31.

Reply:

Thank you for pointing out the errors in the labeling of supplementary Figures 30-31. We have conducted a thorough review of the supplementary materials and corrected the mislabeled figures to ensure their accuracy and consistency.

Response Letter

We sincerely thank the reviewers for valuable comments on our manuscript (Manuscript Number: NCOMMS-24-61933A). We have thoroughly considered the comments and have made substantial revision to the manuscript and supporting information accordingly. Below, we provide a point-to-point response to the reviewers' comments.

Reviewer #1 (Remarks to the Author):

Overall, the authors have done a considerable amount of additional analysis, explanations and writing to improve the paper. The work presents an interesting way of designing a plasma reactor, with a reasonable understanding of the underlying mechanisms (it should be noted that in plasma catalysis a lot of uncertainties and unknowns are present). I find the science in this work very interesting and I now recommend publication of this paper, if the comments related to the figures' clarity below are resolved.

Most figures in the SI are now good with the exception of, at least, Figs. S30, 37, 38, 39, 40, which still display noticeable pixelation. In the main article, all figures still display noticeable pixelation and compression artifacts. My advice is to have a careful look at the settings of the plotting software. Normally, you can configure the absolute figure size and resolution, which should avoid all problems currently present with the figures. Note that it seems to me that in some places the ratio of the (sub)figure also might have changed. Another suggestion for clarity of figure 1c: Please include an arrow pointing to the second y-axis similar as in figure 1h.

Reply:

We appreciate the reviewer's feedback. We have enhanced the quality of Figs. S30, 37, 38, 39, and 40 to reduce pixelation. Additionally, we have added an arrow to indicate the second y-axis in Figure 1c, similar to Figure 1h, for improved clarity.

Supplementary information:

Supplementary Fig. 30 *In situ* FTIR spectra of different catalysts under plasma activation. (a) $\text{Mn}_3\text{O}_4/\text{m-SiO}_2$. (b) $\text{WMO}/\text{m-SiO}_2$. (c) $\text{Na}_2\text{WO}_4/\text{m-SiO}_2$.

Supplementary Fig. 37 DFT-optimized geometries of intermediates in the plasma catalytic conversion of CH_4 to C_2H_2 and C_2H_4 on the Mn_3O_4 (211) surface. Top image (side view) and bottom image (top view) of (a) $^*\text{CH}_4$. (b) $^*\text{CH}_3$. (c) $^*\text{CH}_2$. (d) $^*\text{CH}$. (e) $^*\text{C}_2\text{H}_6$. (f) $^*\text{C}_2\text{H}_4$. (g) $^*\text{C}_2\text{H}_2$. (h) $^*\text{H}$. Element color coding: Mn (slate blue), O (red), C (black) and H (white).

Supplementary Fig. 38 DFT-optimized geometries of intermediates for plasma catalytic conversion of CH_4 to C_2H_2 and C_2H_4 on the Na_2WO_4 (111) surface. Top image (side) and bottom image (top) views of (a) $^*\text{CH}_4$. (b) $^*\text{CH}_3$. (c) $^*\text{CH}_2$. (d) $^*\text{CH}$. (e) $^*\text{C}_2\text{H}_6$. (f) $^*\text{C}_2\text{H}_4$. (g) $^*\text{C}_2\text{H}_2$. (h) $^*\text{H}$. Element color coding: Na-purple, W-dark blue, O-red, C-black and H-white.

Supplementary Fig. 39 DFT-optimized geometries of intermediates in the plasma catalytic conversion of CH_4 to C_2H_2 and C_2H_4 on the $\text{SiO}_2/\text{Mn}_3\text{O}_4$ (211) surface. Top image (side view) and bottom image (top view) of (a) $^*\text{CH}_4$. (b) $^*\text{CH}_3$. (c) $^*\text{CH}_2$. (d) $^*\text{CH}$. (e) $^*\text{C}_2\text{H}_6$. (f) $^*\text{C}_2\text{H}_4$. (g) $^*\text{C}_2\text{H}_2$. Element color coding: Mn (slate blue), Si (yellow), O (red), C (black), H (white).

Supplementary Fig. 40 DFT-optimized geometries of intermediates in the plasma catalytic conversion of CH_4 to C_2H_2 and C_2H_4 on the $\text{SiO}_2/\text{Na}_2\text{WO}_4$ (111) surface. Top image (side view) and bottom image (top view) of (a) $^*\text{CH}_4$. (b) $^*\text{CH}_3$. (c) $^*\text{CH}_2$. (d) $^*\text{CH}$. (e) $^*\text{C}_2\text{H}_6$. (f) $^*\text{C}_2\text{H}_4$. (g) $^*\text{C}_2\text{H}_2$. Element color coding: Na (purple), Si (yellow), W (dark blue), O (red), C (black), H (white).

In the manuscript:

Fig. 1 | Characterization and catalytic performance. **a**, TEM image and particle size distribution of m-SiO₂. **b**, SEM image of WMO/m-SiO₂. **c**, Production rate and molar fraction of C₂H₄ and C₂H₂ within C₂-C₃ hydrocarbons (Conditions: 1 bar, specific energy input (SEI) 5.1 kJ L⁻¹, where SEI is defined as plasma discharge power divided by the gas flow rate; Feed gas 5 vol% CH₄/Ar, total flow rate 200 mL min⁻¹, discharge power 17 W, experiment duration 60 min). **d-e**, TEM and HRTEM images of WMO/m-SiO₂. **f**, EDS line scans (from Point 1 to Point 2) from Fig. 1(d). **g**, Pore size distributions of m-SiO₂, WMO/m-SiO₂, WMO/ZSM-5 and WMO/SiO₂. **h**, Production rate of C₂-C₃ hydrocarbons and methane conversion for catalysis-only, plasma-only and plasma-catalysis systems.